# RACE Attention: A Strictly Linear-Time Attention Layer for Training on Outrageously Large Contexts

**Sahil Joshi     Agniva Chowdhury     Amar Kanakamedala**
**Ekam Singh     Evan Tu     Anshumali Shrivastava**
Department of Computer Science
Rice University
Houston, TX 77005, USA
`{sj157,ac508,ask20,es100,et62,as143}@rice.edu`

## Abstract

Softmax Attention has a quadratic time complexity in sequence length, which becomes prohibitive to run at long contexts, even with highly optimized GPU kernels. For example, FlashAttention-2/3 (exact, GPU-optimized implementations of Softmax Attention) cannot complete a single forward–backward pass of a single attention layer once the context exceeds $\sim 4$ million tokens on an NVIDIA GH200 (96 GB). We introduce **R**epeated **A**rrays-of-**C**ount **E**stimators (RACE) Attention, a kernel-inspired alternative to Softmax Attention that is strictly linear in sequence length and embedding size. RACE Attention replaces the exponential kernel with a sharpened angular similarity, and approximates attention outputs via Gaussian random projections and *soft* Locality-Sensitive Hashing (LSH), avoiding construction of the full attention matrix. Across language modeling, masked language modeling, and text/image classification, RACE Attention matches or outperforms strong baselines up to 64K sequence length while reducing wall-clock time and memory usage. In addition, we conduct a controlled scaling study on a single attention layer and demonstrate processing of up to 12 million tokens on an NVIDIA GH200 GPU and 75 million tokens on an Intel Xeon® Gold 5220R CPU in a single forward–backward pass, which is well beyond the capabilities of current state-of-the-art attention implementations. RACE Attention thus offers a practical and theoretically grounded mechanism for long-context training on today's hardware. We release our code at https://github.com/sahiljoshi515/RACE_Attention.

## 1 Introduction

The Transformer (Vaswani et al., 2017; Dehghani et al., 2019) is the backbone of modern sequence modeling across language, vision (Parmar et al., 2018), and speech (Luo et al., 2020). We have seen remarkable improvements over the past few years in reasoning and understanding capabilities. Most of these are attributed to the increased parameters of the transformers along with the capability to process longer context windows than before. All this progress, however, rests on a computationally expensive primitive: Softmax Attention, whose time scales quadratically with context length. As models and contexts grow, spanning multi-document reasoning, long-form code, audio, and video, the quadratic barrier increasingly dictates who can train and deploy capable systems. Industrial labs mitigate the cost with large-scale distributed hardware; most practitioners cannot. There is a growing need for attention mechanisms that are *accurate*, *fast*, and *memory-efficient*. To highlight the limits of Softmax Attention: even with FlashAttention-2/3 (Dao, 2024; Shah et al., 2024), the state-of-the-art GPU implementations, a single forward–backward pass of a single attention layer (1 batch, 4 heads, 128 embedding size) remains computationally and memory intensive and cannot process sequences beyond $\sim 4$ million tokens on an NVIDIA GH200 (96 GB) GPU. Clearly, to achieve an outrageously long context where the target context size is hundreds of millions of tokens or beyond, fundamental rethinking of attention (Bahdanau et al., 2014) will be required.

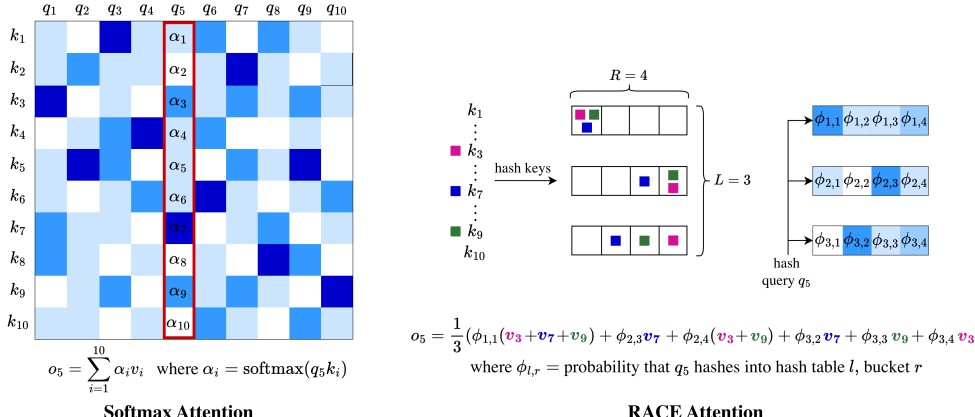

Figure 1: This figure demonstrates the difference between the linear complexity of RACE Attention and the quadratic complexity of Softmax Attention. Specifically, we highlight how the final representation $o_5$ is computed under Softmax versus RACE. In Softmax, the entire fifth column of the attention score matrix is required. In contrast, RACE does not require the full matrix; instead, it aggregates statistics within LSH-mapped buckets, utilizing the appropriate collision probability $\alpha$ to compute $o_5$.

**Linearized and Low-Rank Approximations to Quadratic Attention:** Due to the significance of the problem, a very large body of work attempts to accelerate attention by approximating softmax with linear approximations (Zeng et al., 2021) or clever kernel feature maps (Katharopoulos et al., 2020; Choromanski et al., 2021; Peng et al., 2021; Qin et al., 2022). A work more closely related to ours is YOSO Attention (Zeng et al., 2021), which also approximates a powered angular kernel using hard LSH. Despite this same kernel formulation, the underlying estimators differ substantially. RACE Attention leverages a smooth, differentiable relaxation of classical RACE-based hashing (Coleman & Shrivastava, 2020) to approximate the powered angular kernel. More importantly, YOSO lacks formal theoretical guarantees on its approximation quality and a mechanism that enables causal language modeling. RACE addresses both of these limitations: Sections 3.2 and 3.3 develop a theoretically grounded estimator with quantifiable approximation error, and Algorithm 2 presents a practical attention mechanism that naturally supports causal settings. We defer an in depth comparison between RACE and YOSO to Section 3.1. Two notable lines of work in the direction of linearly approximating attention are Linear Attention (Katharopoulos et al., 2020) and Performer (Choromanski et al., 2021). Linear Attention replaces the softmax similarity with a simple positive kernel via a feature map, e.g. $\phi(x) = elu(x) + 1$. This lets the attention be reordered into associative sums, achieving linear computation. Although such a kernel trick reduces computational complexity, it often degrades accuracy, as clearly demonstrated by our experiments in Section 4. Performer takes a different approach and cleverly leverages the classical idea of approximating the exponential of an inner product using Random Fourier Features (Rahimi & Recht, 2007). However, this strategy comes with its own drawbacks. In particular, the method incurs a time complexity that is quadratic in the embedding size, which offsets many of the intended computational savings. Furthermore, it is well established that approximations based on Random Fourier Features require high-dimensional representations to achieve satisfactory accuracy (Backurs et al., 2017). Our experimental results in fig. 4 reinforce this limitation by showing that these methods exhibit poor scalability in practice.

Another category of work replaces the full $N \times N$ attention matrix with a low-rank surrogate. Some methods learn length-wise projections for keys/values (e.g., *Linformer*), while others use Nyström landmarks to approximate Softmax Attention matrix with a rank-$k$ decomposition (e.g., *Nyströmformer*). These approaches reduce the leading cost from $O(N^2 d)$ to $O(Nkd)$, at the cost of choosing (and occasionally increasing) $k$ to maintain accuracy (Wang et al., 2020; Xiong et al., 2021). Moreover, these methods provide no support for autoregressive tasks. As shown in Section 4, our method outperforms Linformer in accuracy despite Linformer having 13% more parameters than the other methods. Beyond the empirical shortcomings, a deeper conceptual issue persists: existing

approximation approaches lack a rigorous mathematical framework to characterize the trade-offs between efficiency and accuracy. For example, while Performer provides strong kernel-approximation guarantees, a general framework connecting efficiency knobs (e.g., feature count $m$) to downstream accuracy remains limited, and strong accuracy frequently entails large $m$ in practice. As a result, design decisions often appear ad hoc and fragile, leaving methods vulnerable to instability between tasks and settings. Taken together, these limitations explain why, despite the abundance of approximations, Softmax Attention continues to remain the most widely adopted and reliable formulation.

**Sparsity is Complementary:** We note that there is also a popular line of work (Beltagy et al., 2020; Zaheer et al., 2020; Kitaev et al., 2020; Han et al., 2024; Petrick et al., 2022) that exploits structural information in natural language, with sparsity in attention being among the most widely studied. These approaches are complementary to our proposal, which focus on making the attention mechanism itself more efficient and mathematically grounded. In principle, our method can be integrated with structural priors such as sparsity to further improve scalability and accuracy. However, since our objective in this paper is to develop fundamentally efficient attention, we will not discuss this line of structural approaches further, instead we view combining them with our method as an important direction for future work.

**Key Idea:** Standard attention relies on the well-known softmax function, computing

$$O = \text{softmax}\left(\frac{QK^\top}{\sqrt{d}}\right) V \,, \tag{1}$$

where the softmax is applied row-wise so that attention weights are nonnegative and sum to one. In this paper, we leverage an alternative to the softmax—namely, a higher-degree monomial of an *angular* kernel based on cosine geometry:

$$O = \left(1 - \left(\frac{\cos^{-1}(QK^\top)}{\pi}\right)\right)^\gamma V \tag{2}$$

Eq. 2 should be read as an *informal* analogue to Eq. 1, where the angular kernel replaces the exponential. A more precise definition, with explicit cosine normalization and row-wise normalization, is given in Section 3.1. We argue that, for sufficiently large values of $\gamma$, this formulation closely mimics the behavior of softmax and refer to it as *Angular Attention* in the subsequent sections. Importantly, it admits a linear-time approximation algorithm. In particular, we leverage the connection (Section 2.2) between **R**epeated **A**rrays-of-**C**ount **E**stimators (RACE) (Coleman & Shrivastava, 2020; Luo & Shrivastava, 2018) and the angular kernel to design our algorithm in Section 3.2. We therefore refer to our proposed method as *RACE Attention*.

RACE Attention is a drop-in replacement for Softmax Attention. We evaluate it in Transformers on causal language modeling, masked language modeling, and text/image classification (Section 4 and Appendix A.2). By reframing similarity around a powered angular kernel and using differentiable LSH-based sketches, it provides a principled alternative that supports very long contexts on commodity hardware. The sketching view keeps constant factors small: each query mixes with only a fixed bank of $S = LR$ bucket summaries rather than all $N$ keys. Since we never materialize the full attention matrix, the working set stays compact and activation memory drops, enabling much longer sequences with reduced latency. In contrast, FlashAttention-2/3 (Dao, 2024; Shah et al., 2024) reduce the memory footprint of attention via tiling, but still require computing all key-query interactions, preventing processing of much longer sequences at comparable speed, as shown in fig. 4. In addition to our novel findings about RACE Attention and rigorous supporting experimental evidence, we provide the following:

**I. Long-context scaling:** RACE scales to sequence lengths far beyond prior attention mechanisms, processing up to *75M tokens* on CPU and *12M tokens* on GPU in a single forward–backward pass of a single attention layer, using the same hyperparameters as in our accuracy evaluations.

**II. Trainable RACE:** We introduce a differentiable sketch by replacing hard hashing with smooth soft assignments over hypercube corners, enabling end-to-end training.

**III. CPU/GPU pre-training:** Our approach supports both *causal* (autoregressive) and *non-causal* (bidirectional) pre-training on CPU and GPU, with custom OpenMP/CUDA kernels that compute causal prefix operations in a single streaming pass for linear-time, memory-efficient training.

**IV. Theoretical Insights:** Section 3.3 establishes approximation guarantees based on the LSH framework and analyzes how the parameters $L$ (number of hash tables) and $R$ (buckets per table) govern the variance–accuracy tradeoff.

## 2 BACKGROUND

### 2.1 LOCALITY-SENSITIVE HASHING (LSH)

An LSH family $\mathcal{H}$ for a similarity Sim makes near pairs collide more often than far pairs. Formally, $\mathcal{H}$ is $(S_0, cS_0, p_1, p_2)$-sensitive if for all $x, y \in \mathbb{R}^D$,

$$\begin{cases} \mathrm{Sim}(x,y) \geq S_0 \;\Rightarrow\; \mathrm{Pr}_{h\sim\mathcal{H}}[h(x) = h(y)] \geq p_1, \\ \mathrm{Sim}(x,y) \leq cS_0 \;\Rightarrow\; \mathrm{Pr}_{h\sim\mathcal{H}}[h(x) = h(y)] \leq p_2, \end{cases}$$

where $p_1 > p_2$ and $c < 1$. Such families enable sublinear-time approximate nearest-neighbor data structures. A convenient sufficient condition, satisfied by SimHash and WTA (Charikar, 2002; Yagnik et al., 2011; Chen & Shrivastava, 2018), is that the collision probability is a monotone function of similarity, $\mathrm{Pr}_{h\sim\mathcal{H}}[h(x) = h(y)] = f(\mathrm{Sim}(x,y))$ with $f$ increasing.

### 2.2 RACE SKETCH

RACE (Coleman & Shrivastava, 2020; Coleman et al., 2020) shows that any similarity expressible as a (non-negative) linear combination of LSH collision kernels can be sketched using ACE-style estimation (Luo & Shrivastava, 2018). It provides an *unbiased* estimator of kernel-density sums for LSH collision kernels and their *powers*. In particular, RACE estimates $\sum_{x\in D} k(x,q)^p$ by hashing items into counters and reading the counters addressed by the query; averaging across $L$ independent rows reduces variance.

**Lemma 1** (Theorem 1 of (Coleman & Shrivastava, 2020))**.** *Given a dataset $D$, an LSH family $H$ with finite range $[1, R]$ and a parameter $p$, construct an LSH function $h(x) \to [1, R^p]$ by concatenating $p$ independent hashes from $H$. Let $A$ be an ACE array constructed using $h(x)$. Then for any query $q$,*

$$\mathbb{E}\big[A[h(q)]\big] \;=\; \sum_{x\in D} k(x,q)^p$$

## 3 INTRODUCING RACE ATTENTION

---

**Algorithm 1** RACE Attention (non-causal)

---

**Input:** $Q, K, V \in \mathbb{R}^{N\times d}$; number of hash tables $L$; number of hyperplanes $P$; temperature $\beta > 0$.
**Output:** $\widehat{O} \in \mathbb{R}^{N\times d}$.

1: **for** $\ell = 1, \ldots, L$ **do**
2:      Draw $W^{(\ell)} \in \mathbb{R}^{P\times d}$ with rows $w_p^{(\ell)} \overset{\text{i.i.d.}}{\sim} \mathcal{N}(0, I_d)$.
3:      Define the corner set $\mathcal{V} = \{\pm 1\}^P$ $(R = 2^P)$ with $v_r \in \{\pm 1\}^P$.
4:      Build $\Phi_Q^{(\ell)}, \Phi_K^{(\ell)} \in \mathbb{R}^{N\times R}$ with rows

$$[\phi^{(\ell)}(x)]_r = \frac{\exp\{\beta\,(\tanh(W^{(\ell)}x))^\top v_r\}}{\sum_{r'}\exp\{\beta\,(\tanh(W^{(\ell)}x))^\top v_{r'}\}}, \quad x \in \{Q_i, K_j\}$$

5:      Per-table bucket statistics:

$$A^{(\ell)} \;=\; (\Phi_K^{(\ell)})^\top \mathbf{1}_N \in \mathbb{R}^R, \qquad B^{(\ell)} \;=\; (\Phi_K^{(\ell)})^\top V \in \mathbb{R}^{R\times d}.$$

6: **end for**
7: Compute average across tables: $\texttt{Num} \;=\; \frac{1}{L}\sum_{\ell=1}^L \Phi_Q^{(\ell)} B^{(\ell)}$ and $\texttt{Den} \;=\; \frac{1}{L}\sum_{\ell=1}^L \Phi_Q^{(\ell)} A^{(\ell)}$.
8: Return $\widehat{O} \leftarrow \mathrm{diag}(\texttt{Den})^{-1}\,\texttt{Num}$

---

### 3.1 SOFTMAX-LIKE SIMILARITIES THAT ADMIT LINEAR-TIME ESTIMATION

Given a sequence of $N$ tokens, a Transformer produces for each position $i$ a query $Q_i \in \mathbb{R}^d$, and for every position $j$ a key $K_j \in \mathbb{R}^d$ and a value $V_j \in \mathbb{R}^d$.

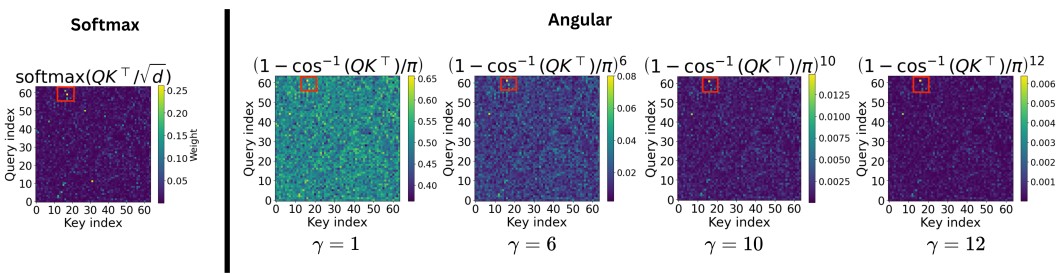

Figure 3: Comparison of Softmax and Angular kernels at different sharpening levels $\gamma$. As $\gamma$ (non-linearity) increases, Angular transitions from flat similarity scores to a sharper distribution, recovering behavior similar to the exponential in Softmax.

The output at position $i$ is a weighted sum of the values, where the weight on $V_j$ reflects the relevance of token $j$ to token $i$. In the standard formulation (Vaswani et al., 2017), relevance is computed via the scaled dot product given by Eq. 1. This choice guarantees two useful properties of the attention weights: (i) non-negativity and (ii) they sum to one, so $O_i$ is a convex combination of the values. Equally important, the exponential introduces a strong non-linear mapping from similarity scores to attention weights, amplifying small score differences. This observation suggests a broader view: attention weights can be derived from any normalized highly non-linear (exponential like) similarity function. Let $\text{sim} : \mathbb{R}^d \times \mathbb{R}^d \to \mathbb{R}_{\geq 0}$ be any non-negative similarity function. We can define normalized *similarity attention* as

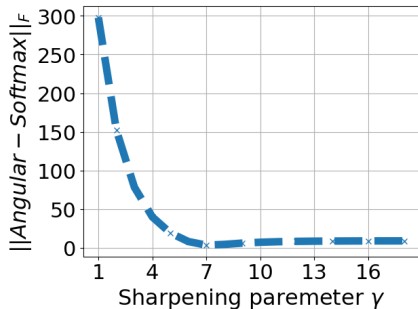

Figure 2: Frobenius error between Angular and Softmax Attention vs. $\gamma$.

$$O_i = \frac{\sum_{j=1}^{N} \text{sim}(Q_i, K_j)\, V_j}{\sum_{j=1}^{N} \text{sim}(Q_i, K_j)} \tag{3}$$

Our quest is for finding non-linear (softmax-like) similarity kernels that admit accurate linear-time estimation, eliminating the quadratic cost of attention in both training and inference. Similar to YOSO (Zeng et al., 2021), we argue that a good starting point is a well known LSHable (Coleman & Shrivastava, 2020; Choromanski et al., 2017) *angular* similarity. It is well behaved and normalized, in particular, it depends only on the angle between the vectors $Q_i$ and $K_j$ and is invariant to their norms: $\text{sim}(Q_i, K_j) = 1 - \cos^{-1}\left(\frac{Q_i^\top K_j}{\|Q_i\| \|K_j\|}\right)/\pi$. However, unlike exponential in softmax, the raw angular kernel is relatively flat near high similarity values, reducing its ability to sharply discriminate between nearly aligned vectors. To increase contrast, we propose to exponentiate the angular kernel with a sharpening parameter $\gamma$, which accentuates differences among highly similar pairs. After sharpening the kernel, the similarity function becomes as follows:

$$\text{sim}(Q_i, K_j) = \left(1 - \cos^{-1}\left(\frac{Q_i^\top K_j}{\|Q_i\| \|K_j\|}\right)/\pi\right)^\gamma \tag{4}$$

In fig. 3, we show that for sufficiently large $\gamma$, the angular kernel becomes almost indistinguishable from softmax kernel. This is expected because a higher degree monomial like $x^{12}$ behaves similarly to an exponential. Furthermore, in fig. 2 we plot the frobenius error between Angular and Softmax Attention. The error sharply decreases as $\gamma$ increases, demonstrating that softmax-level sharpness can be achieved with modest polynomial degree (*e.g.*, $\gamma = 8$).

In its current form, evaluating the attention with similarity given by Eq. 4 is no different from softmax. It naively still requires all $N^2$ query-key interactions. Fortunately, any constant exponentiation of angular kernel, belongs to a family, that admits efficient kernel density estimation using RACE sketches (Coleman & Shrivastava, 2020), and we use these sketches to approximate the kernel in linear time obtaining an algorithmically efficient alternative to Softmax Attention!

Finally, as outlined in Section 1, YOSO (Zeng et al., 2021) and our method operate on the same kernel in Eq. 4, but differ fundamentally in how attention is computed. YOSO employs hard LSH to generate Bernoulli collision indicators, yielding an unbiased estimator of the unnormalized attention numerator $\sum_{j=1}^{N} \text{sim}(Q_i, K_j)V_j$ in Eq. 3. It then applies a post-hoc $\ell_2$ normalization, which does not correspond to standard attention normalization. Since this hard formulation is non-differentiable, YOSO relies on surrogate lower-bound gradients derived from additional Bernoulli samples rather than directly differentiating the kernel. This introduces quadratic complexity in the embedding dimension $d$, resulting in poor scalability during end-to-end training (see fig. 5c). To see how RACE constructs a smooth, differentiable LSH-based estimator that is linear in $d$ in contrast to YOSO, refer to Section 3.2.

## 3.2 THE FINAL ALGORITHM

At a high level, RACE avoids explicitly approximating the $N \times N$ attention matrix, which would otherwise remain quadratic. Instead, RACE sketches the sufficient statistics needed to compute attention outputs directly in linear time. Fig. 1 illustrates this difference using $o_5$, the output embedding of token 5. In Softmax Attention, computing $o_5$ requires the full attention column, whereas RACE softly assigns queries and keys (Qs and Ks) into $R$ LSH-indexed bucket summaries under the same LSH scheme. The values are weighted by the key probabilities and then combined with the query probabilities. Averaging over $L$ independent hash tables further reduces variance, following standard sketching practice. Comprehensive visualizations are provided in Appendix (figs. 7, 8).

We next formalize the RACE Attention mechanism in Algorithm 1. As described in Section 2.2, one final technical hurdle remains: the RACE algorithm is non-differentiable. We get around the non-differentiability of RACE sketches by replacing discrete bucket assignments with soft probabilities and using standard cross-entropy loss, preserving differentiability for end-to-end training. Algorithm 1 consists of three key stages: (i) **Soft bucketization:** Each query/key $x \in \mathbb{R}^d$ is randomly projected via $W^{(\ell)}$ hyperplanes and softly assigned to $R = 2^P$ corners with distribution $\phi^{(\ell)}(x)$ (steps 2–4), (ii) **Bucket aggregation:** For each table $\ell$, we form per-bucket statistics by accumulating key weights and their weighted values, namely the mass vector $A^{(\ell)} \in \mathbb{R}^R$ and the value-sum matrix $B^{(\ell)} \in \mathbb{R}^{R \times d}$, so that $A^{(\ell)}[r]$ is the total (soft) mass in bucket $r$ and $B^{(\ell)}[r,:]$ is the corresponding sum of values. (step 5), (iii) **Global normalization:** The algorithm averages across $L$ tables to form $\texttt{Num} = \frac{1}{L}\sum_\ell \Phi_Q^{(\ell)} B^{(\ell)}$ and $\texttt{Den} = \frac{1}{L}\sum_\ell \Phi_Q^{(\ell)} A^{(\ell)}$, and reconstructs the final outputs as $\widehat{O} = \text{diag}(\texttt{Den})^{-1}\texttt{Num}$ (steps 7–8).

A useful way to interpret Algorithm 1 is through the kernel perspective introduced in Section 2.2. In classical RACE (Coleman & Shrivastava, 2020), $P$ random hyperplanes generate a hash $h(x) = \text{sign}(W^{(\ell)}x)$, and two vectors collide under this hash with probability $\Pr[h(Q_i) = h(K_j)] = S_{ij} := \text{sim}(Q_i, K_j)$, which is exactly the $P$-powered angular kernel in Eq. 4 with $\gamma = P$. In soft RACE, we keep this geometric structure intact, but replace the hard sign map with a smooth approximation: we compute a "soft" sign vector $\tanh(W^{(\ell)}x)$ and evaluate its alignment with each of the $R = 2^P$ corner sign patterns $v_r \in \{\pm 1\}^P$, assigning $x$ to buckets with nonzero probability via a softmax over these alignments. This turns the discrete collision event of classical RACE into a differentiable quantity while preserving its underlying angular dependence. In particular, vectors with small angular distance still assign most of their mass to the same buckets, mirroring the behavior of the $P$-powered angular kernel. As a result, the per-table quantity $\phi^{(\ell)}(Q_i)^\top \phi^{(\ell)}(K_j)$ serves as a smooth approximation to the $P$-powered angular similarity that our attention mechanism seeks to approximate. We formalize these kernel quantities in the Section 3.3, where the RACE-based approximation $\widehat{S}_{ij}$ is introduced.

**Computational Complexity:** The per-table runtime of Algorithm 1 can be decomposed according to its main steps: Step 2 (random projections) costs $\mathcal{O}(NdP)$, Step 3 (logits over $R = 2^P$ corners) costs $\mathcal{O}(NPR)$, and Step 5 (bucket aggregation) costs $\mathcal{O}(NRd)$. The global accumulation in Step 7 adds $\mathcal{O}(NRd)$ *per table*. Thus, the per-table runtime is $\mathcal{O}(NdP + NPR + NRd) = \mathcal{O}(NRd)$, with memory $\mathcal{O}(NR + Rd)$. Across $L$ tables, this becomes $\mathcal{O}(LNRd)$ time and $\mathcal{O}(L(NR + Rd))$ space. Compared to FlashAttention-2/3's $\mathcal{O}(N^2d)$ time and $\mathcal{O}(Nd)$ space, RACE is more efficient since $R, L \ll N$ and $R, L \ll d$, even for moderate $N$ and $d$.

### 3.3 THEORETICAL ANALYSIS OF ALGORITHM 1

Algorithm 1 is presented in terms of random projections, soft bucketization, and per-bucket aggregation. For analysis it is easier to take a *kernel approximation* view of attention. Each hash table $\ell = 1, \ldots, L$ induces a randomized feature map $\phi^{(\ell)} : \mathbb{R}^d \to \mathbb{R}^R$, where $R = 2^P$ is the number of hypercube corners, and defines the approximate kernel $\widehat{S}_{ij}^{(\ell)} = \left(\phi^{(\ell)}(Q_i)\right)^\top \phi^{(\ell)}(K_j)$. Then, averaging across $L$ independent tables yields $\widehat{S} = \frac{1}{L} \sum_{\ell=1}^{L} \widehat{S}^{(\ell)}$. This view places soft RACE Attention in the language of kernel methods: it replaces the angular kernel ($\gamma = P$) in Eq. 4 with the randomized sketch $\widehat{S}$ based on LSH-style features. Since $\phi^{(\ell)}(x)$ is a softmax distribution, the approximate kernel $\widehat{S}$ inherits concentration properties from the underlying random Gaussian projections. This allows us to analyze its deviation from the target angular kernel using standard tools from randomized numerical linear algebra (RandNLA) (Tropp, 2015). Our analysis requires the following two mild assumptions on the target kernel $S$:
**(A1)** Row sums of $S$ are bounded away from zero *i.e.,* $s_{\min} := \min_i (S\mathbf{1})_i \geq C_1 N$ for some constant $C_1 > 0$, which ensures stable normalization in attention.
**(A2)** Spectral norm of $S$ is bounded *i.e.,* $\|S\|_2 \leq C_2 N$, which follows from $S_{ij} \in [0, 1]$.

Several comments are necessary to better understand the above structural conditions. Condition (A1) rules out degenerate cases where a query has vanishing similarity with all keys, which would make the row-normalization in attention unstable. In practice this assumption is mild: with learned representations, attention rows rarely collapse to near-zero mass, so requiring $s_{\min} \geq C_1 N$ simply rules out degenerate cases where a query is effectively isolated (assigns negligible weight to almost all keys). Condition (A2) is even less restrictive: since $S_{ij} \in [0, 1]$, the worst case is attained by the all-ones matrix $J_N$, whose spectral norm is exactly $\|J_N\|_2 = N$. Thus bounding $\|S\|_2 \leq C_2 N$ merely rules out pathological growth beyond this trivial maximum, and is always satisfied with $C_2 = 1$. We are now ready to state our main quality-of-approximation result:

**Theorem 2.** *Let $Q, K, V \in \mathbb{R}^{N \times d}$ be the query, key, and value matrices. For parameters $L$, $P$, and $\beta$, and under conditions (A1) and (A2), the estimator $\widehat{O}$ produced by Algorithm 1 satisfies*

$$\|\widehat{O} - O\|_{\mathrm{rms}} \;=\; \mathcal{O}\left(\frac{P}{\beta} + \sqrt{\frac{\log(N/\delta)}{L}}\right) \|V\|_F$$

*with probability at least $1 - \delta$. Here, $O \in \mathbb{R}^{N \times d}$ with the $i^{th}$ row $O_i$ is defined using Eqs. equation 3 and equation 4 with $\gamma = P$, and $\|\widehat{O} - O\|_{\mathrm{rms}} := \sqrt{\frac{1}{N} \sum_{i=1}^{N} \|\widehat{O}_i - O_i\|_2^2}$ denotes per-token root-mean-square (RMS) error between $O$ and $\widehat{O}$.*

The bound decomposes into a *bias term* $\mathcal{O}(P/\beta)$ and a *variance term* $\mathcal{O}(\sqrt{\log(N/\delta)/L})$. Larger $\beta$ reduces the bias, while increasing $L$ reduces the variance. The dependence on $P$ arises because powering the angular kernel by $P$ makes collisions sharper, but soft bucketization (finite $\beta$) smooths out these decisions and introduces an additional bias. To keep this bias small, $\beta$ should be scaled with $P$. In particular, as $\beta, L \to \infty$, the approximation error vanishes. In fact, taking $L = \Theta(\log N)$ prevents the variance from exploding. Together, this kernel reinterpretation provides a precise RandNLA lens for analyzing *RACE Attention*, with $L$, $P$, and $\beta$ jointly governing its accuracy-efficiency trade-offs. The proof of Theorem 2, together with all intermediate lemmas, is deferred to Appendix B due to space constraints.

**Remark (Causal masking):** Our language modeling experiments (Tables 4 and 8) use causal RACE Attention, implemented efficiently in OpenMP/CUDA (Algorithm 2). Our theory, however, covers only the non-causal setting. Extending Theorem 2's bias-variance guarantees to the causal case remains an open problem, since the cumulative-sum constraint interacts non-trivially with the random-feature construction. A rigorous causal analysis is a key direction for future work.

## 4 EXPERIMENTS

To ensure comparability and avoid cherry-picking, we adopt the standard evaluation suites used in prior efficient-attention work—Linear Attention, Linformer, and Random Feature Attention (RFA) (Katharopoulos et al., 2020; Wang et al., 2020; Peng et al., 2021). We evaluate **text classification** on QNLI, SST-2, IMDB, Yahoo, and Arxiv (Rajpurkar et al., 2016; Socher et al., 2013;

Table 1: Long-context classification performance on a 40GB A100 GPU. Train/Test denote per-epoch runtimes in seconds, and Acc. denotes accuracy.

| | Arxiv Long-Document Classification | | | | | | | | |
|---|---|---|---|---|---|---|---|---|---|
| | 16K | | | 32K | | | 64K | | |
| Method | Train ↓ | Test ↓ | Acc. ↑ | Train ↓ | Test ↓ | Acc. ↑ | Train ↓ | Test ↓ | Acc. ↑ |
| RACE (P=2,L=2) | **80.5s** | 3.9s | 70.3% | **282s** | 15s | 89.4% | **561s** | 22s | 97.14% |
| RACE (P=3,L=3) | 82.4s | 4.0s | **71.3%** | 289s | 15.6s | 90.6% | 584s | 22.5s | 97.92% |
| RACE (P=4,L=4) | 84.7s | 4.1s | 70.8% | 300s | 16s | **91.1%** | 594s | 22.9s | 97.4% |
| Linear | 83.8s | 4.0s | 67.9% | 286s | 15.9s | 87.3% | 591s | 22.8s | 96.35% |
| Linformer-128 | 86s | **3.2s** | 64.1% | 296s | **10.7s** | 87.5% | 616s | **15.2s** | 97.4% |
| Performer-256 | 128s | 5.8s | 68.9% | 449s | 24.6s | 86.5% | 952s | 35s | 96.61% |
| FlashAttention2 | 95.7s | 3.7s | 69.8% | 471s | 20s | 89.7% | 1645s | 47s | 97% |

Table 2: **Comparison of attention methods across diverse tasks.**

| Method | CIFAR-10 @1024 Acc. (%) ↑ | QNLI @2048 Acc. (%) ↑ | Tiny Stories @1024 PPL ↓ |
|---|---|---|---|
| RACE (P=2, L=2) | 63.7 | 60.7 | 4.2 |
| RACE (P=3, L=3) | 62.5 | 60.7 | 3.2 |
| RACE (P=4, L=4) | **65.9** | 61.1 | 2.6 |
| Linear | 60.0 | 60.7 | 7.0 |
| Linformer-128 | 63.7 | 60.6 | 3.7 |
| FlashAttention2 | 61.44 | 61.1 | 2.7 |
| Angular ($\gamma=8$) | 61.69 | **61.7** | **2.5** |
| Performer-256 | 64.9 | 61.0 | 10.0 |
| Sigmoid | 57.2 | 61.1 | 3.7 |

Table 3: **Food-101 @ 16K (Image Classification)**

| Method | Train ↓ | Test ↓ | Acc. ↑ |
|---|---|---|---|
| RACE (P=2, L=2) | **891s** | **37s** | 42.4% |
| RACE (P=3, L=3) | 950s | 40s | **43.5%** |
| RACE (P=4, L=4) | 1042s | 42s | 40.3% |
| Linear | 1166s | 44s | 41.4% |
| Linformer-128 | 1250s | 49s | 20.2% |
| Performer-256 | 2546s | 105s | 42.4% |
| FlashAttention2 | 2600s | 95s | 42.1% |

Maas et al., 2011; Zhang et al., 2015; He et al., 2019); **autoregressive language modeling** on WikiText-103 and PTB (Merity et al., 2017; Marcus et al., 1993); **masked language modeling** on Tiny Stories (Eldan & Li, 2023); **image classification** on CIFAR-10, FashionMNIST, and Food-101 using ViT architecture (Krizhevsky, 2009; Xiao et al., 2017; Bossard et al., 2014; Dosovitskiy et al., 2021); and **long-context reasoning** on Long Range Arena tasks (e.g., ListOps and Text Retrieval) (Tay et al., 2021). Together, these benchmarks span bidirectional and autoregressive modeling, as well as moderate- and long-context text and image classification. During training, we treat $\beta$, introduced in Algorithm 1, as a trainable parameter.

Beyond standard benchmarks, we conduct extreme-length scaling experiments on a single attention layer, reaching sequences of tens of millions of tokens while using the same hyperparameters as in the accuracy evaluations. Finally, FlashAttention-2/3 are exact fused implementations of Softmax Attention, and we use the terms interchangeably when reporting accuracy and runtime.

**Baselines:** We compare RACE against widely used baselines with publicly available implementations: FlashAttention2 (Dao et al., 2022), Linear Attention (Katharopoulos et al., 2020), Performer (Choromanski et al., 2021), Linformer (Wang et al., 2020), Sigmoid Attention (Ramapuram et al., 2025), and YOSO (Zeng et al., 2021). These span exact, kernel-linear, and low-rank approximations. All models are tuned per authors' guidelines and trained under identical settings.

**Is RACE Attention as Accurate as Transformers?** We report **text-classification**, **image-classification**, and **masked language modeling** results in Tables 1, 2, and 3. Specifically, in Table 3, "Train/Test" denote per-epoch runtimes in seconds, and "Acc." denotes accuracy. Results for Linear Attention, Linformer, and Performer were collected using batch size 1 due to out-of-memory failures at batch size 8. In contrast, both RACE and FlashAttention2 remain memory-efficient at this sequence length and are evaluated with batch size 8. Additional experiments can be found in Appendix, refer to Tables 6, 7, 9, and 10. For **autoregressive language modeling**, RACE matches softmax-level perplexity on WikiText-103 and improves upon it on PTB (Tables 4 and 8). These results indicate that RACE preserves accuracy in the overlapping regime while delivering consistent gains on moderate- and long-context settings. Unless stated otherwise, all methods use the same Transformer backbone (layers, heads, embedding size, dropout) and training budget.

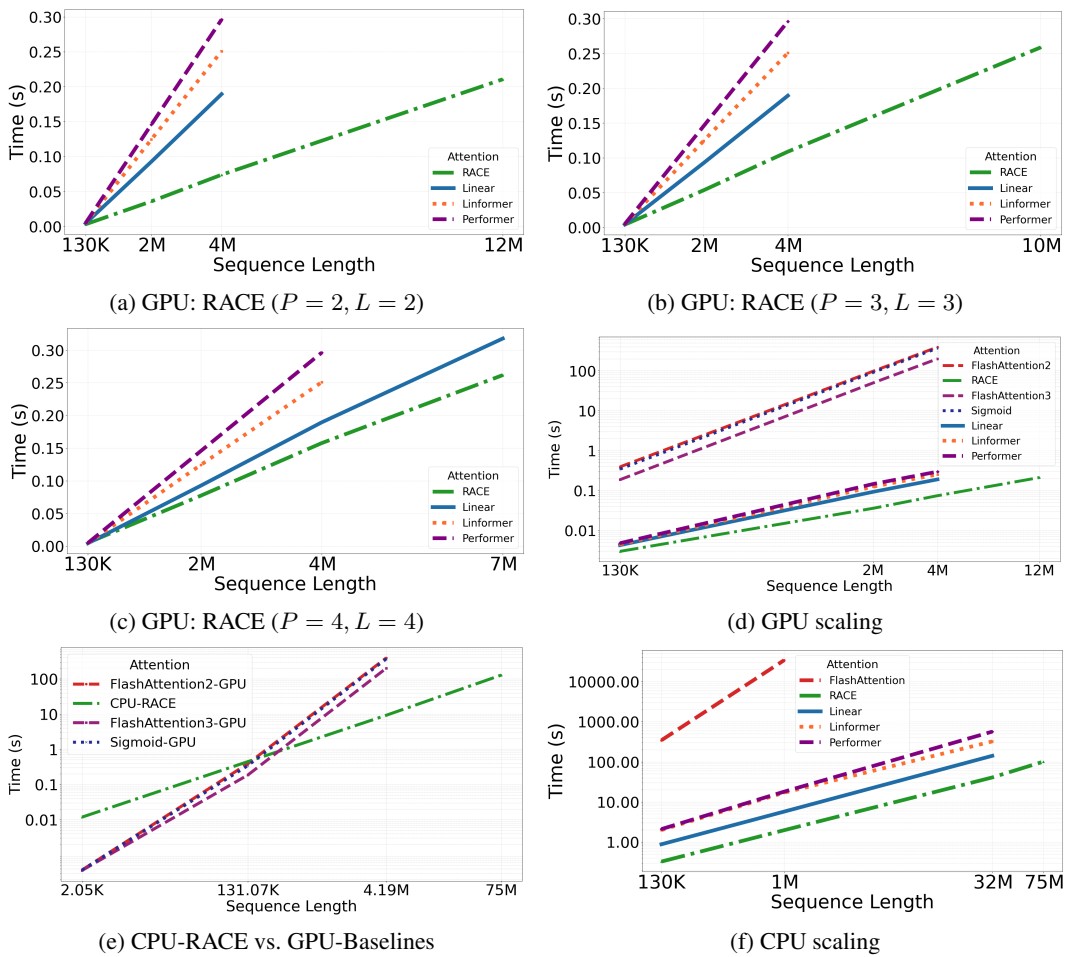

(a) GPU: RACE ($P = 2, L = 2$)

(b) GPU: RACE ($P = 3, L = 3$)

(c) GPU: RACE ($P = 4, L = 4$)

(d) GPU scaling

(e) CPU-RACE vs. GPU-Baselines

(f) CPU scaling

Figure 4: **Scaling stress-tests on GPU and CPU:** All plots use logarithmic axes and report a single forward–backward pass (batch size 1, 4 attention heads, $d$=128) of a single attention layer. Linformer and Performer use the same hyperparameters as in Table 2. (a–d) compare the scaling behavior of RACE against attention baselines on GPU, while (e–f) show scaling for RACE ($P$=3, $L$=3) on CPU, highlighting its superior efficiency at long sequence lengths.

We train with identical optimizers, schedulers, and batch sizes; full hyperparameters appear in Table 5. Metrics are reported from the best-validation checkpoint. All accuracy experiments were conducted on a single NVIDIA A100 GPU.

**Can we reach 100 million context window on popular hardware?** Now, we evaluate how RACE Attention scales across common hardware relative to strong baselines. For RACE, we use sketch parameters $(P, L)$ chosen to match FlashAttention2's accuracy/perplexity on the same tasks in Table 2. For each method, we measure the wall-clock time for a single forward–backward pass of a single attention layer with 1 batch, 4 heads, and embedding size 128, as a function of sequence length, stress-testing

Table 4: **WikiText-103 @ 1024**

| Method | PPL $\downarrow$ |
|---|---|
| RACE (P=2, L=2) | 23.9 |
| RACE (P=2, L=3) | 23.4 |
| RACE (P=3, L=3) | 21.9 |
| RACE (P=3, L=4) | 21.5 |
| RACE (P=4, L=4) | 20.9 |
| FlashAttention2 | 20.9 |
| Angular ($\gamma$=8) | **19.0** |

context lengths up to 100 million tokens. Since FlashAttention-2/3 are designed specifically for GPU hardware, we use PyTorch's optimized `F.scaled_dot_product_attention` implementation as the FlashAttention baseline when reporting CPU scaling results.

**How far can we scale attention on standard Intel Xeon® Gold 5220R CPU?** RACE scales to sequences of up to **75 million tokens** for a single forward–backward pass of a single attention layer on CPU. By contrast, FlashAttention becomes prohibitively slow at $\sim 2$ million tokens due to the

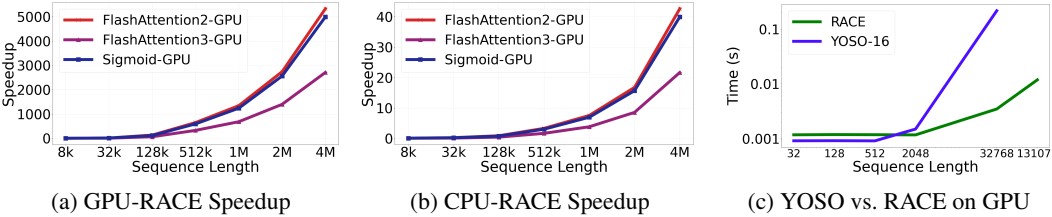

(a) GPU-RACE Speedup        (b) CPU-RACE Speedup        (c) YOSO vs. RACE on GPU

Figure 5: Speedup of RACE relative to GPU-only attention baselines, with RACE executed on GPU in (a) and on CPU in (b). (c) YOSO (CUDA kernel) fails beyond 32K tokens due to memory constraints, whereas RACE continues to scale to much longer sequences.

quadratic time scaling in sequence length $N$ (see fig. 4f). It is worth noting that FlashAttention does not run out of memory on the CPU DRAM. RACE is more than $10000\times$ faster than FlashAttention at context length of $\sim 33$ million. RACE finishes comfortably under 10 seconds for a single forward-backward pass on this hardware while FlashAttention takes approximately $10^5$ seconds on the same hardware. Although the absolute runtime grows with $N$, RACE's speedup over FlashAttention increases. At 75 million tokens, RACE finishes in about 100 seconds. This is expected because RACE is linear and FlashAttention is quadratic in $N$. The experiments also highlight that linear attentions' approximations are not only inaccurate but also significantly slower and have large memory overheads due to large hidden constants (see fig. 6 in Appendix). They run about an order of magnitude slower than RACE Attention and even go out of memory at $\sim 33$ million tokens.

**How far can we scale attention on the most powerful GH200 GPU?** An NVIDIA GH200 has 96GB of memory. Here, we observe a similar trend. RACE scales up to **12 million** tokens for a single forward-backward pass of a single attention layer, whereas FlashAttention-2/3 and Sigmoid Attention becomes impractical around $\sim 4$ million tokens (see fig. 4). At $\sim 4$ million tokens RACE takes merely 0.1 seconds to finish, while FlashAttention2 needs about 550 seconds, making RACE about $5500\times$ faster on GPUs for processing 4 million tokens. Additionally, RACE is about $5000\times$ faster than Sigmoid and $2600\times$ faster than FlashAttention3 for processing 4 million tokens (see fig. 5a). FlashAttention removes the quadratic memory cost of attention scores, but exact attention still requires storing token-level $Q, K, V$, and $O$ activations and their gradients, incurring $\mathcal{O}(BHNd)$ memory. For large $N$, this linear footprint alone exceeds GPU HBM capacity. In contrast, RACE compresses queries and keys into $R$ bucket summaries per table, reducing activation memory to $\mathcal{O}(BHL(NR + Rd))$, which scales more favorably on GPU even than the linear attention baselines that exhaust memory at approximately $\sim 4$ million tokens (see fig. 4). As a result, RACE supports contexts up to $3.5\times$ longer than FlashAttention-2/3 and Sigmoid Attention.

**Right Algorithm beats Hardware Acceleration:** While GPUs offer substantial speedups for a fixed algorithm, comparing FlashAttention-2/3 and Sigmoid on a high-end GH200 GPU against RACE on a single CPU highlights the impact of *algorithmic* acceleration. Fig. 4e reports the runtime of a single forward–backward pass of a single attention layer as the context length increases. For short to moderate sequences ($N \lesssim 131$K), the GPU's massive parallelism dominates and FlashAttention-2/3 remain faster. Beyond this scale, however, the asymptotic behavior becomes decisive: the GPU kernels begin to saturate, and RACE becomes faster. At a sequence length of $\sim 4$ million tokens (the largest supported by FlashAttention-2/3 in our configuration), *RACE on CPU* is roughly $40\times$ faster than FlashAttention-2 and Sigmoid Attention on GPU, and about $20\times$ faster than FlashAttention-3 (see fig. 5b). These results show that, in the long-context regime, state-of-the-art GPU attention kernels are fundamentally constrained by their quadratic dependence on sequence length. In contrast, RACE's algorithmic efficiency enables it to outperform even the most advanced GPU-accelerated methods by a wide margin, despite executing on substantially weaker hardware.

## 5 CONCLUSION

We introduce RACE Attention, a linear-time, memory-efficient alternative to Softmax Attention that approximates a sharpened angular kernel via RACE sketches. This formulation will enable substantially more efficient key–value caching during inference and suggests a clear path toward optimized CUDA kernels that preserve favorable scaling behavior and theoretical guarantees in autoregressive settings. We leave a full exploration of these directions to future work.

## ACKNOWLEDGMENTS

We would like to thank the anonymous reviewers for their helpful comments. We are also grateful to Aditya Desai and Zhaozhuo Xu for their valuable insights and thought-provoking discussions. This work was supported by the National Science Foundation (NSF) CCF-2336612 and Rice Ken Kennedy Institute (K2I) Generative AI Cluster Funding.

## ETHICS STATEMENT

Our work is focused on speeding up algorithm and reducing the memory complexity for Attention. As such, it could have significant broader impacts by allowing practitioners to train models fast and deploy them in constrained resource settings. Our experimental work uses publicly available datasets to evaluate the performance of our algorithm; no ethical considerations are raised.

## REPRODUCIBILITY STATEMENT

We provide the source code and configuration for the key experiments (Language Modelling, Masked Language Modelling, and Classification) including instructions on how to generate data and train the models. All proofs are stated in the appendix with explanations and underlying assumptions. We thoroughly checked the implementation and also verified empirically that the proposed algorithm holds.

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

# A  APPENDIX

## A.1  LLM USAGE

An LLM was used exclusively to refine the language and improve clarity across the manuscript. It was not used for idea generation, experimental design, analysis, or to produce results.

## A.2  ADDITIONAL NOTES ON EXPERIMENTS

Table 5: Experiment Setup and Hyperparameters

| Dataset | Task | Hyperparameters |
|---|---|---|
| PTB | Language Modeling | $N$=128; layers=1; heads=2; $d$=128; batch=16; lr=$6e^{-4}$; $\beta$'s=(0.9, 0.999); $\epsilon$=$1e^{-8}$; wd=0.1; dropout = 0.3; epochs=70 |
| WikiText-103 | Language Modeling | $N$=1024; layers=8; heads=8; $d$=512; batch=16; lr=$6e^{-4}$; $\beta$'s=(0.9, 0.999); $\epsilon$=$1e^{-8}$; wd=0.1; dropout=0.1; epochs=100 |
| IMDB | Text Classification | $N$=512; layers=1; heads=2; $d$=128; batch=32; lr=$1e^{-5}$; wd=$5e^{-5}$; dropout=0.1; epochs=150 |
| Yahoo | Text Classification | N=256; layers=1; heads=2; d=128; batch=32; lr=$1e^{-5}$; wd=$5e^{-5}$; dropout = 0.1; epochs=100 |
| ListOps | Text Classification | $N$=2000; layers=8; heads=8; $d$=512; batch=16; lr=$1e^{-5}$; wd=$1e^{-5}$; dropout=0.1; epochs=40 |
| Text Retrieval | Text Classification | $N$=8000; layers=4; heads=2; $d$=384; batch=1; lr=$2e^{-4}$; wd=$1e^{-2}$; dropout=0.1; epochs=20 |
| Tiny Stories | Masked Language Modelling | $N$=512-1024; layers=6; heads=4; $d$=384; batch=32; lr=$6e^{-4}$; $\beta$'s=(0.9, 0.999); $\epsilon$=$1e^{-8}$; wd=0.1; dropout=0.1; epochs=150; stories=20000 |
| QNLI | Text Classification | $N$=2048; layers=4; heads=8; $d$=384; batch=32; lr=$1e^{-5}$; wd=$5e^{-5}$; dropout=0.1; epochs=100 |
| SST-2 | Text Classification | $N$=1024; layers=4; heads=8; $d$=384; batch=32; lr=$1e^{-5}$; wd=$5e^{-5}$; dropout=0.1; epochs=100 |
| CIFAR-10 | Image Classification | $N$=1024; layers=2; heads=4; $d$=384; batch=32; lr=$6e^{-4}$; $\beta$'s=(0.9, 0.999); $\epsilon$=$1e^{-8}$; wd=0.1; dropout=0.1; epochs=75 |
| FashionMNIST | Image Classification | $N$=784; layers=2; heads=4; $d$=384; batch=32; lr=$6e^{-4}$; $\beta$'s=(0.9, 0.999); $\epsilon$=$1e^{-8}$; wd=0.1; dropout=0.1; epochs=75 |
| Food-101 | Image Classification | $N$=16384; layers=8; heads=8; $d$=512; batch=8; lr=$3e^{-4}$; wd=0.001; dropout=0.1; epochs=100; grad_accum_steps=4 |
| Arxiv | Text Classification | $N$=16K-64K; layers=4; heads=4; $d$=256; batch=2; lr=$3e^{-4}$; wd=0.01; dropout=0.1; epochs=100; grad_accum_steps=16 |

**Description:** Unless otherwise specified, we adopt a *linear warmup–decay learning-rate schedule*. Let $T$ denote the total number of optimizer updates,

$$T = \text{epochs} \times \text{len(train\_loader)}.$$

The learning rate increases linearly from 0 to the base learning rate (reported in Table 5) over the first $0.01T$ updates, and then decays linearly to 0 over the remaining $T - 0.01T$ updates. The learning-rate scheduler is stepped once per optimizer update. We use the AdamW optimizer with hyperparameters listed in Table 5. For experiments with a learning rate below $2 \times 10^{-4}$, we do not apply linear warmup or a learning-rate scheduler; instead, we train the model using a constant learning rate for the number of epochs specified in Table 5.

**Data Pre-processing for Arxiv:** To evaluate long-context classification performance, we construct 16K-, 32K-, and 64K-token variants of the ArXiv dataset using a two-stage preprocessing pipeline. First, we tokenize all documents with a basic English tokenizer and retain only those whose raw token length exceeds a predefined minimum threshold, ensuring that each example contains sufficiently long context. Next, we perform streaming sequence packing. Documents are grouped by class and concatenated sequentially until a target sequence length of 16K, 32K, or 64K tokens is reached. To maintain high packing efficiency, only very small residual fragments are discarded. This procedure yields long, contiguous sequences that preserve the natural structure of individual documents while enabling fixed-length training. After balancing classes, the resulting dataset sizes are as follows: 16K (1,947 training examples, 209 test examples), 32K (3,650 training examples, 520 test examples), and 64K (3,882 training examples, 384 test examples).

Table 6: **FashionMNIST @ 784**

| Method | Accuracy |
|---|---|
| RACE (P=2, L=5) | **87.7%** |
| RACE (P=3, L=5) | 87.5% |
| RACE (P=4, L=4) | 86.6% |
| RACE (P=4, L=5) | 85.7% |
| Linformer-128 | **87.7%** |
| FlashAttention2 | 87.2% |
| Angular ($\gamma$=8) | 86.4% |
| Linear | 85.8% |
| Performer-256 | 86.6% |

Table 7: **Tiny Stories @ 512**

| Method | Perplexity |
|---|---|
| RACE (P=3, L=4) | 3.9 |
| RACE (P=4, L=4) | 3.3 |
| RACE (P=5, L=4) | **2.7** |
| RACE (P=5, L=5) | 5.1 |
| Linear | 6.0 |
| Angular ($\gamma$=8) | 2.9 |
| FlashAttention2 | 3.1 |
| Linformer-128 | 4.6 |
| Performer-256 | 7.1 |

Table 8: **Penn Tree Bank @ 128**

| Method | Test PPL |
|---|---|
| RACE (P=2, L=2) | 54.7 |
| RACE (P=3, L=3) | 54.2 |
| RACE (P=4, L=4) | **53.4** |
| Angular ($\gamma$=8) | 58.8 |
| Angular ($\gamma$=12) | 57.6 |
| Linear | 73.2 |
| FlashAttention2 | 55.4 |

Table 9: Comparison of attention methods on Long Range Arena

| Method | ListOps @2000 Acc. (%) ↑ | Text Retrieval @8000 Acc. (%) ↑ |
|---|---|---|
| RACE (P=2, L=2) | 41.9% | 80.3% |
| RACE (P=3, L=3) | 41.3% | 80.8% |
| RACE (P=4, L=4) | 41.6% | **80.9%** |
| Linformer-128 | 38.9% | 76.1% |
| FlashAttention2 | 41.4% | 80.5% |
| Angular ($\gamma$=8) | **42.2%** | OOM |
| Linear | 39.6% | 80.6% |
| Performer-256 | 40.2% | 80.8% |
| YOSO-16 | 41.0% | 80.2% |

Table 10: Comparison of attention methods on natural language understanding tasks

| Method | Yahoo @256 Acc. (%) ↑ | IMDB @512 Acc. (%) ↑ | SST-2 @1024 Acc. (%) ↑ |
|---|---|---|---|
| RACE (P=2, L=2) | 66.9% | 80.6% | 76.7% |
| RACE (P=3, L=3) | 66.6% | **81.3%** | 77.1% |
| RACE (P=4, L=4) | 67.2% | **81.3%** | **79.4%** |
| Linformer-128 | 64.7% | 78.2% | 75.1% |
| FlashAttention2 | **67.2%** | 80.0% | 78.5% |
| Angular ($\gamma$=8) | 67.0% | 79.6% | 77.2% |
| Linear | 66.9% | 80.9% | 78.0% |
| Performer-256 | 64.9% | 81.0% | 77.3% |

(a) CPU: RACE ($P=2, L=2$)  (b) CPU: RACE ($P=3, L=3$)  (c) CPU: RACE ($P=4, L=4$)

(d) $d$ scaling @ $N$=65536  (e) GPU: RACE ($P=4, L=4$)

Figure 6: (a–c) CPU results for different $(P, L)$. (d) Linear attention scales quadratically in $d$, while RACE scales linearly (fixed $N$=65536). (e) GPU scaling under hyperparameters different from fig. 4.

**Data Pre-processing for Food-101:**   To evaluate long-context image classification on the Food-101 dataset with a 16K token context length, we reduce the dataset size by discarding 50 classes. We train the model on 20,000 samples and evaluate it on a held-out test set of 2,500 samples. Images are processed using a patch size of 4, an input resolution of $512 \times 512$, and a stride of 4. All remaining hyperparameters are reported in Table 5.

# B PROOF OF THEOREM 2

This section provides a complete, self-contained theoretical treatment showing that our RACE Attention closely approximates Angular Attention. We give explicit high-probability bounds for (i) the kernel error, (ii) the attention matrix error with a clean separation of numerator vs. denominator effects, and (iii) the end-to-end output error (Theorem 2). Let's also reintroduce the notations for the convenience of the reader.

## B.1 SETUP AND ASSUMPTIONS

**Data.** Sequence length $N$, head (per-head) dimension $d$. Queries/keys are unit vectors:

$$Q_i, K_j \in \mathbb{R}^d \quad \text{with} \quad \|Q_i\|_2 = \|K_j\|_2 = 1, \quad i, j \in \{1, \ldots, N\}.$$

**Target kernel ($P$-powered angular).**

$$\kappa(Q_i, K_j) := \kappa_{\mathrm{ang}}(Q_i, K_j)^P = \left(1 - \tfrac{1}{\pi} \cos^{-1}(Q_i^\top K_j)\right)^P \in [0, 1], \qquad S \in \mathbb{R}^{N \times N} \text{ with } S_{ij} = \kappa(Q_i, K_j).$$

**Soft RACE features.** For each ensemble $\ell = 1, \ldots, L$:

- Draw $P$ random hyperplanes $W^{(\ell)} \in \mathbb{R}^{P \times d}$ whose rows $w_t^{(\ell)}$ are i.i.d.
- Corners $\mathcal{V} = \{\pm 1\}^P$ (size $R = 2^P$), with corner vectors $v_r \in \{\pm 1\}^P$.
- Logits $s^{(\ell)}(x; r) := [\tanh(W^{(\ell)} x)]^\top v_r$, temperature $\beta > 0$.
- Define the (probability) feature $\phi^{(\ell)}(x)$ by

$$[\phi^{(\ell)}(x)]_r = \frac{\exp\{\beta\, s^{(\ell)}(x; r)\}}{\sum_{r'} \exp\{\beta\, s^{(\ell)}(x; r')\}}.$$

**RACE kernel and matrices.** For each ensemble, define the per-table kernel matrix

$$\widehat{S}_{ij}^{(\ell)} = (\phi^{(\ell)}(Q_i))^\top (\phi^{(\ell)}(K_j)), \qquad \widehat{S} = \frac{1}{L} \sum_{\ell=1}^{L} \widehat{S}^{(\ell)}.$$

Let the (single-table) bias matrix be $\tilde{B} := \mathbb{E}[\widehat{S}^{(\ell)}] - S$.

**Assumptions.** For convenience, we restate the two assumptions from Section 3.3

- **(A1)** Row sums of $S$ are bounded away from zero *i.e.,* $s_{\min} := \min_i (S\mathbf{1})_i \geq C_1 N$ for some constant $C_1 > 0$, which ensures stable normalization in attention.
- **(A2)** Spectral norm of $S$ is bounded *i.e.,* $\|S\|_2 \leq C_2 N$, which follows from $S_{ij} \in [0, 1]$.

**Notation:** We denote $\|\cdot\|_2$ as spectral norm for a matrix and Euclidean norm for a vector, $\|\cdot\|_F$ for the Frobenius norm of a matrix and for a matrix M, we denote $\|M\|_\infty = \max_i \sum_j |M_{ij}|$, is the induced row-sum norm.

## B.2 KERNEL CONSTRUCTION WITH THE BIAS TERM

We begin by formalizing how a single hash table induces a kernel matrix via the soft RACE features. The next lemma records norm properties that will be used repeatedly.

**Lemma 3** (Bounds for a single ensemble)**.** *Let $\Phi_Q^{(\ell)} \in \mathbb{R}^{N \times R}$ be the matrix with the $i$-th row $\phi^{(\ell)}(Q_i)^\top$ and $\Phi_K^{(\ell)}$ defined analogously. Then:*

1. *$\widehat{S}^{(\ell)} = \Phi_Q^{(\ell)} (\Phi_K^{(\ell)})^\top$.*

2. *Each row of $\Phi_Q^{(\ell)}$ and $\Phi_K^{(\ell)}$ is a probability vector; hence $\|\Phi_Q^{(\ell)}\|_F, \|\Phi_K^{(\ell)}\|_F \leq \sqrt{N}$.*

3. *Consequently $\|\widehat{S}^{(\ell)}\|_F \leq N$.*

*Proof.* Each $\phi^{(\ell)}(x)$ is a softmax over $R = 2^P$ corners, so entries are nonnegative and sum to 1. Item (1) is by definition of $\widehat{S}_{ij}^{(\ell)}$. For (2), every row $p$ satisfies $\|p\|_2 \leq \|p\|_1 = 1$, hence $\|\Phi_Q^{(\ell)}\|_F^2 = \sum_i \|\phi^{(\ell)}(Q_i)\|_2^2 \leq N$ (and similarly for $\Phi_K^{(\ell)}$). Item (3) follows from $\|AB\|_F \leq \|A\|_F \|B\|_F$. $\square$

Having controlled the feature-induced matrix norms, we quantify the zero-mean fluctuation of one ensemble around its expectation and prepare moment bounds needed for matrix concentration. Note that the hash projections $W^{(\ell)}$ (and hence $\widehat{S}^{(\ell)}$) are independent for $\ell = 1, \ldots, L$.

**Lemma 4.** *Let $X^{(\ell)} := \widehat{S}^{(\ell)} - \mathbb{E}[\widehat{S}^{(\ell)}]$ and write*

$$\Delta := \widehat{S} - S = \frac{1}{L} \sum_{\ell=1}^{L} X^{(\ell)} + \tilde{B}.$$

*Then:*

1. $\mathbb{E}[X^{(\ell)}] = 0$.

2. $\|X^{(\ell)}\|_2 \leq 2N$.

3. *With*

$$v := \max \left\{ \left\| \sum_{\ell=1}^{L} \mathbb{E}\left[ \left( \tfrac{1}{L} X^{(\ell)} \right) \left( \tfrac{1}{L} X^{(\ell)} \right)^\top \right] \right\|_2, \left\| \sum_{\ell=1}^{L} \mathbb{E}\left[ \left( \tfrac{1}{L} X^{(\ell)} \right)^\top \left( \tfrac{1}{L} X^{(\ell)} \right) \right] \right\|_2 \right\},$$

*we have $v \leq 4N^2/L$.*

*Proof.* **(1)** By definition, $X^{(\ell)} = \widehat{S}^{(\ell)} - \mathbb{E}[\widehat{S}^{(\ell)}]$, hence $\mathbb{E}[X^{(\ell)}] = \mathbb{E}[\widehat{S}^{(\ell)}] - \mathbb{E}[\widehat{S}^{(\ell)}] = 0$.

**(2)** By Lemma 3(3) we have $\|\widehat{S}^{(\ell)}\|_2 \leq \|\widehat{S}^{(\ell)}\|_F \leq N$. By convexity of the spectral norm,

$$\|\mathbb{E}[\widehat{S}^{(\ell)}]\|_2 \leq \mathbb{E}\big[\|\widehat{S}^{(\ell)}\|_2\big] \leq N.$$

Therefore, by the triangle inequality,

$$\|X^{(\ell)}\|_2 = \|\widehat{S}^{(\ell)} - \mathbb{E}[\widehat{S}^{(\ell)}]\|_2 \leq \|\widehat{S}^{(\ell)}\|_2 + \|\mathbb{E}[\widehat{S}^{(\ell)}]\|_2 \leq 2N.$$

**(3)** Let $Y^{(\ell)} := \frac{1}{L} X^{(\ell)}$. Then

$$\sum_{\ell=1}^{L} \mathbb{E}\big[Y^{(\ell)}(Y^{(\ell)})^\top\big] = \frac{1}{L^2} \sum_{\ell=1}^{L} \mathbb{E}\big[X^{(\ell)}(X^{(\ell)})^\top\big].$$

Using subadditivity of $\|\cdot\|_2$, Jensen, and $\|AB\|_2 \leq \|A\|_2 \|B\|_2$,

$$\left\| \sum_{\ell=1}^{L} \mathbb{E}\big[Y^{(\ell)}(Y^{(\ell)})^\top\big] \right\|_2 \leq \frac{1}{L^2} \sum_{\ell=1}^{L} \left\| \mathbb{E}\big[X^{(\ell)}(X^{(\ell)})^\top\big] \right\|_2 \leq \frac{1}{L^2} \sum_{\ell=1}^{L} \mathbb{E}\big[\|X^{(\ell)}\|_2^2\big] \leq \frac{1}{L^2} \sum_{\ell=1}^{L} (2N)^2 = \frac{4N^2}{L}.$$

The same bound holds for $\left\| \sum_{\ell=1}^{L} \mathbb{E}\big[(Y^{(\ell)})^\top Y^{(\ell)}\big] \right\|_2$ by symmetry. Taking the maximum of the two yields $v \leq 4N^2/L$. $\square$

To convert the moment and uniform bounds above into a high-probability spectral-norm bound, we invoke a standard matrix Bernstein inequality from Tropp (2015), stated next for completeness.

**Lemma 5** (Matrix Bernstein). *If $Z^{(\ell)} \in \mathbb{R}^{m \times n}$ are independent mean-zero matrices with $\|Z^{(\ell)}\|_2 \leq H$ and variance proxy $v$, then for any $t > 0$,*

$$\mathbb{P}\left( \left\| \sum_\ell Z^{(\ell)} \right\|_2 \geq t \right) \leq (m + n) \exp\left( -\frac{t^2/2}{v + Ht/3} \right).$$

Next, applying Lemma 5 with the parameters established in Lemma 4, we obtain the following nonasymptotic deviation bound for the kernel estimator.

**Theorem 6** (Kernel deviation with explicit constants). *With probability at least $1 - \delta$,*

$$\|\widehat{S} - S\|_2 \ \leq \ \|\tilde{B}\|_2 \ + \ 4\frac{N}{\sqrt{L}}\sqrt{\log\frac{2N}{\delta}} \ + \ \frac{4}{3}\frac{N}{L}\log\frac{2N}{\delta}.$$

*Proof.* First, rewrite $\widehat{S} - S$ as

$$\widehat{S} - S = \frac{1}{L}\sum_{\ell=1}^{L}\widehat{S}^{(\ell)} - S = \frac{1}{L}\sum_{\ell=1}^{L}\left(\widehat{S}^{(\ell)} - \mathbb{E}[\widehat{S}^{(\ell)}]\right) \ + \ \left(\mathbb{E}[\widehat{S}^{(\ell)}] - S\right) = \frac{1}{L}\sum_{\ell=1}^{L}X^{(\ell)} + \tilde{B}.$$

By the triangle inequality,

$$\|\widehat{S} - S\|_2 \ \leq \ \left\|\frac{1}{L}\sum_{\ell=1}^{L}X^{(\ell)}\right\|_2 \ + \ \|\tilde{B}\|_2.$$

It remains to upper bound the random term with high probability.

Set $Z^{(\ell)} := \frac{1}{L}X^{(\ell)}$. Then the $Z^{(\ell)}$ are independent, mean-zero, $N \times N$ random matrices. From Lemma 4(2) we have $\|X^{(\ell)}\|_2 \leq 2N$. Therefore, $\|Z^{(\ell)}\|_2 \leq H := \frac{2N}{L}$. Similarly, Lemma 4(3) gives $v \leq \frac{4N^2}{L}$. Applying Lemma 5 with $m = n = N$ yields

$$\mathbb{P}\left(\left\|\sum_{\ell=1}^{L}Z^{(\ell)}\right\|_2 \geq t\right) \ \leq \ 2N\exp\left(-\frac{t^2}{2\left(v + Ht/3\right)}\right).$$

Let $u := \log\frac{2N}{\delta}$. To make the RHS $\leq \delta$, it suffices that

$$\frac{t^2}{2\left(v + Ht/3\right)} \ \geq \ u \quad \Longleftrightarrow \quad t^2 - \frac{2uH}{3}t - 2uv \ \geq \ 0.$$

Choose

$$t \ = \ 2\sqrt{v\,u} \ + \ \frac{2}{3}H\,u.$$

Writing $a := 2\sqrt{vu}$ and $b := \frac{2}{3}Hu$ (so $t = a + b$) gives

$$t^2 - \frac{2uH}{3}t - 2uv = (a + b)^2 - \frac{2uH}{3}(a + b) - 2uv = (4vu - 2uv) + \left(\tfrac{8}{3} - \tfrac{4}{3}\right)Hu\sqrt{vu} \geq 0.$$

Therefore,

$$\left\|\sum_{\ell=1}^{L}Z^{(\ell)}\right\|_2 \ \leq \ 2\sqrt{v\,u} \ + \ \frac{2}{3}H\,u \quad \text{with probability at least } 1 - \delta.$$

Plugging $v \leq \frac{4N^2}{L}$ and $H = \frac{2N}{L}$ yields

$$\left\|\sum_{\ell=1}^{L}Z^{(\ell)}\right\|_2 \ \leq \ 2\sqrt{\frac{4N^2}{L}u} \ + \ \frac{2}{3}\cdot\frac{2N}{L}u = 4\frac{N}{\sqrt{L}}\sqrt{u} \ + \ \frac{4}{3}\frac{N}{L}u.$$

Since $\sum_{\ell=1}^{L}Z^{(\ell)} = \frac{1}{L}\sum_{\ell=1}^{L}X^{(\ell)}$, we conclude that

$$\left\|\frac{1}{L}\sum_{\ell=1}^{L}X^{(\ell)}\right\|_2 \ \leq \ 4\frac{N}{\sqrt{L}}\sqrt{\log\frac{2N}{\delta}} \ + \ \frac{4}{3}\frac{N}{L}\log\frac{2N}{\delta} \quad \text{with probability } \geq 1 - \delta,$$

and therefore

$$\|\widehat{S} - S\|_2 \ \leq \ \|\tilde{B}\|_2 \ + \ 4\frac{N}{\sqrt{L}}\sqrt{\log\frac{2N}{\delta}} \ + \ \frac{4}{3}\frac{N}{L}\log\frac{2N}{\delta},$$

as claimed. $\qquad\qquad\square$

The deviation bound decomposes into a variance term and a (deterministic) bias term $\tilde{B}$. We now bound $\tilde{B}$ explicitly as a function of $\beta$ and $P$. Before stating the result, we introduce the following technical result that proves a deterministic inequality and will be required to bound $\|\tilde{B}\|_2$.

**Lemma 7.** *Let $p, q \in \mathbb{R}^R$ be probability vectors with nonnegative entries summing to one. Let $a \in \arg\max_r p_r$ and $b \in \arg\max_r q_r$. Then*

$$\left| p^\top q - \mathbf{1}\{a = b\} \right| \leq (1 - p_a) + (1 - q_b).$$

*Proof.* We consider two cases.

**Case 1:** $a = b$. Then $\mathbf{1}\{a = b\} = 1$ and

$$|p^\top q - 1| = 1 - p^\top q \leq 1 - p_a q_a \quad \text{since } p^\top q \geq p_a q_a.$$

By the inequality $(1 - x)(1 - y) \geq 0 \Rightarrow 1 - xy \leq (1 - x) + (1 - y)$, we obtain

$$1 - p^\top q \leq (1 - p_a) + (1 - q_a) = (1 - p_a) + (1 - q_b).$$

**Case 2:** $a \neq b$. Then $\mathbf{1}\{a = b\} = 0$. Because $b$ maximizes $q$, we have $q_r \leq q_b$ for all $r$ and $q_a \leq 1 - q_b$ (since the total mass outside $b$ is $1 - q_b$). Hence

$$p^\top q = p_a q_a + \sum_{r \neq a} p_r q_r \leq p_a(1 - q_b) + q_b \sum_{r \neq a} p_r = p_a(1 - q_b) + (1 - p_a)q_b.$$

Finally,

$$(1 - p_a) + (1 - q_b) - \left[ p_a(1 - q_b) + (1 - p_a)q_b \right] = 2(1 - p_a)(1 - q_b) \geq 0,$$

so $p^\top q \leq (1 - p_a) + (1 - q_b)$.

Combining both cases gives the desired bound. $\qquad\qquad\square$

**Lemma 8** (Bounding the bias term). *Fix $P \geq 1$ and $\beta > 0$. With $S$ and $\tilde{B}$ as above, let $c := 2\tanh(1)$. Then*

$$\|\tilde{B}\|_2 \leq \frac{4}{\sqrt{2\pi}} \frac{NP}{\beta} + \left( \frac{4}{\sqrt{2\pi}} e^{-1/2} \right) NP e^{-c\beta}.$$

*Proof.* Let us denote the inner product similarity by $\rho := Q_i^\top K_j$, and recall that the angular kernel is $\kappa_{\mathrm{ang}}(Q_i, K_j) := 1 - \frac{1}{\pi} \cos^{-1}(\rho)$ From standard LSH theory, for $P$ i.i.d. Gaussian hyperplanes $W^{(\ell)} \in \mathbb{R}^{P \times d}$, the probability that all $P$ bits match is exactly:

$$\mathbb{P}\left( h_P(Q_i) = h_P(K_j) \right) = \kappa_{\mathrm{ang}}(Q_i, K_j)^P = S_{ij}.$$

Now define the softmax sketch feature for the hash table $\ell$:

$$s^{(\ell)}(x; r) := \tanh(W^{(\ell)} x)^\top v_r, \qquad [\phi^{(\ell)}(x)]_r := \frac{e^{\beta s^{(\ell)}(x; r)}}{\sum_{r'} e^{\beta s^{(\ell)}(x; r')}},$$

where $v_r \in \{\pm 1\}^P$ denotes the binary corner vectors of length $P$, and $R = 2^P$. Let $\widehat{S}^{(\ell)} \in \mathbb{R}^{N \times N}$ be the kernel matrix for a single hash table:

$$\mathbb{E}(\widehat{S}_{ij}^{(\ell)}) := \mathbb{E}\left[ \left( \phi^{(\ell)}(Q_i) \right)^\top \left( \phi^{(\ell)}(K_j) \right) \right], \qquad S_{ij} := \kappa_{\mathrm{ang}}(Q_i, K_j)^P,$$

and recall the bias matrix is $\tilde{B} := \mathbb{E}[\widehat{S}^{(\ell)}] - S$. Our goal is to bound $\|\tilde{B}\|_2$. To do this, fix any pair $(i, j)$ and note:

$$|\mathbb{E}(\widehat{S}_{ij}^{(\ell)}) - S_{ij}| = \left| \mathbb{E}\left[ \left( \phi^{(\ell)}(Q_i) \right)^\top \left( \phi^{(\ell)}(K_j) \right) \right] - \mathbb{P}[h_P(Q_i) = h_P(K_j)] \right|.$$

Next, let $r^\star(x) = \arg\max_r s^{(\ell)}(x; r) = \mathrm{sign}(u(x))$ denote the maximizing corner for $x$, where $u_t(x) = \tanh(w_t^\top x)$. The function $s^{(\ell)}(x; r) = \sum_{t=1}^P u_t(x) r_t$ is a linear form over the binary

corners $r \in \{\pm 1\}^P$. To evaluate the normalization term in the softmax, note that the exponentials factorize across coordinates because $r_t$ appears only in the term $u_t(x)r_t$. Hence,

$$\sum_{r \in \{\pm 1\}^P} e^{\beta s^{(\ell)}(x;r)} = \sum_{r_1 = \pm 1} \cdots \sum_{r_P = \pm 1} \left( e^{\beta \sum_t u_t(x)r_t} \right) = \sum_{r_1 = \pm 1} \cdots \sum_{r_P = \pm 1} \prod_{t=1}^{P} e^{\beta u_t(x)r_t}$$

$$= \prod_{t=1}^{P} \left( \sum_{r_t = \pm 1} e^{\beta u_t(x)r_t} \right) = \prod_{t=1}^{P} \left( e^{\beta u_t(x)} + e^{-\beta u_t(x)} \right) = \prod_{t=1}^{P} 2\cosh(\beta |u_t(x)|),$$

where the last equality uses the evenness of the hyperbolic cosine, $\cosh(z) = \cosh(|z|)$. Therefore, the softmax probability assigned to the dominant corner $r^\star(x)$ can be written in closed form as

$$[\phi^{(\ell)}(x)]_{r^\star(x)} = \frac{e^{\beta \sum_t |u_t(x)|}}{\prod_t 2\cosh(\beta |u_t(x)|)} = \prod_{t=1}^{P} \frac{e^{\beta |u_t(x)|}}{e^{\beta |u_t(x)|} + e^{-\beta |u_t(x)|}} = \prod_{t=1}^{P} \sigma\big(2\beta |u_t(x)|\big),$$

where $\sigma(z) = 1/(1+e^{-z})$ denotes the logistic function. This factorization expresses the max-corner mass as a product of per-coordinate logistic terms.

To bound the total probability mass outside the dominant corner, we use the inequality $1 - \prod_t (1 - a_t) \le \sum_t a_t$ for $a_t \in [0,1]$, which we apply to $a_t = 1 - \sigma(2\beta |u_t(x)|)$. Hence,

$$1 - [\phi^{(\ell)}(x)]_{r^\star(x)} = 1 - \prod_{t=1}^{P} \sigma(2\beta |u_t(x)|) \le \sum_{t=1}^{P} \left( 1 - \sigma(2\beta |u_t(x)|) \right) \le \sum_{t=1}^{P} e^{-2\beta |u_t(x)|}, \quad (5)$$

where the final inequality follows from the standard logistic bound $1 - \sigma(z) \le e^{-z}$ for all $z \ge 0$. Intuitively, this means that the total softmax probability mass outside the most likely corner decays exponentially with the scaled activation strength $\beta |u_t(x)|$ along each coordinate. Now, for each bit, $u_t(x) = \tanh(w_t^\top x)$ with $w_t^\top x \sim \mathcal{N}(0,1)$. Let $Z := w_t^\top x$. Then

$$\mathbb{E}[e^{-2\beta |u_t(x)|}] = \mathbb{E}[e^{-2\beta |\tanh(Z)|}].$$

We split into two regions. (1) On $|Z| \le 1$, we use $|\tanh z| \ge \frac{|z|}{2}$, so $e^{-2\beta |\tanh(Z)|} \le e^{-\beta |Z|}$. Hence

$$\mathbb{E}[e^{-2\beta |\tanh(Z)|} \mathbf{1}_{|Z| \le 1}] \le \frac{2}{\sqrt{2\pi}} \int_0^1 e^{-\beta z} e^{-z^2/2} \, dz \le \frac{2}{\sqrt{2\pi}} \cdot \frac{1}{\beta}.$$

(2) On $|Z| > 1$, we use $\tanh z \ge \tanh(1)$, so $e^{-2\beta |\tanh(Z)|} \le e^{-2\beta \tanh(1)}$. Thus

$$\mathbb{E}[e^{-2\beta |\tanh(Z)|} \mathbf{1}_{|Z| > 1}] \le e^{-2\beta \tanh(1)} \mathbb{P}(|Z| > 1) = 2e^{-2\beta \tanh(1)} \mathbb{P}(Z > 1) \le e^{-2\beta \tanh(1)} \sqrt{\frac{2}{\pi}} e^{-1/2} \le e^{-c\beta},$$

where the second last bound is due to Mill's inequality and the last inequality follows from the fact that $\sqrt{\frac{2}{\pi}} e^{-1/2} \le 1$. Here $c = 2\tanh(1)$. Combining the two expectations we get,

$$\mathbb{E}[e^{-2\beta |u_t(x)|}] \le \frac{2}{\sqrt{2\pi}\,\beta} + e^{-c\beta}, \quad c = 2\tanh(1).$$

Substituting back into Eq. equation 5, we obtain

$$\mathbb{E}\big[1 - [\phi^{(\ell)}(x)]_{r^\star(x)}\big] \le \frac{2P}{\sqrt{2\pi}\,\beta} + \mathcal{O}(Pe^{-c\beta}).$$

Next, let $p := \phi^{(\ell)}(Q_i)$ and $q := \phi^{(\ell)}(K_j)$ denote the softmax feature vectors for $Q_i$ and $K_j$, and let $a := r^\star(Q_i)$, $b := r^\star(K_j)$ be their respective dominant corners. By the deterministic inequality in Lemma 7

$$|p^\top q - \mathbf{1}\{a = b\}| \le (1 - p_a) + (1 - q_b),$$

valid for any probability vectors $p, q$ and indices $a, b$, we have

$$|\mathbb{E}[\widehat{S}_{ij}^{(\ell)}] - S_{ij}| = \big|\mathbb{E}[p^\top q] - \mathbb{P}[a = b]\big| \le \mathbb{E}[1 - p_a] + \mathbb{E}[1 - q_b].$$

Using the bound on the expected softmax tail probability for both $Q_i$ and $K_j$ then gives

$$|\mathbb{E}[\widehat{S}_{ij}^{(\ell)}] - S_{ij}| \le 2\left(\frac{2P}{\sqrt{2\pi}\,\beta} + \mathcal{O}(Pe^{-c\beta})\right) = \frac{4P}{\sqrt{2\pi}\,\beta} + \mathcal{O}(Pe^{-c\beta}). \qquad (6)$$

Therefore,

$$\|\tilde{B}\|_2 \le \|\tilde{B}\|_F \le N \sup_{i,j} |\tilde{B}_{ij}| \le N\left(\frac{4P}{\sqrt{2\pi}\,\beta} + \mathcal{O}(Pe^{-c\beta})\right).$$

This proves the claim. $\qquad\qquad\square$

We now propagate the kernel-level error into the attention matrix. This requires controlling the normalization (row sums) and its inverse, which we address next.

### B.3 FROM KERNELS TO ATTENTION (NUMERATOR VS. DENOMINATOR)

Let $\widehat{S} = S + \Delta$, $D = \mathrm{diag}(S\mathbf{1})$, and $\widehat{D} = \mathrm{diag}(\widehat{S}\mathbf{1}) = D + E$. Define the attention matrices

$$A := D^{-1}S, \qquad \widehat{A} := \widehat{D}^{-1}\widehat{S}.$$

The following lemma ties the row-sum perturbation $E$ to $\Delta$ and gives a simple invertibility condition for $\widehat{D}$.

**Lemma 9** (Row-sum and inverse diagonal control). *Recall that $s_{\min} = \min_i D_{ii} > 0$. Then*

1. $\|E\|_2 \le \|\Delta\|_\infty \le \sqrt{N}\,\|\Delta\|_2$.

2. *If $\|E\|_2 \le s_{\min}/2$, then $\|\widehat{D}^{-1}\|_2 \le 2/s_{\min}$.*

*Proof.* **(1) Row-sum control.** Since $\widehat{S} = S + \Delta$ and $\widehat{D} = \mathrm{diag}(\widehat{S}\mathbf{1})$, we rewrite

$$E := \widehat{D} - D = \mathrm{diag}\left((\widehat{S} - S)\mathbf{1}\right) = \mathrm{diag}(\Delta\mathbf{1}).$$

Hence each diagonal entry is $E_{ii} = (\Delta\mathbf{1})_i = \sum_{j=1}^N \Delta_{ij}$, so

$$\|E\|_2 = \max_i |E_{ii}| = \max_i \left|(\Delta\mathbf{1})_i\right| \le \max_i \sum_{j=1}^N |\Delta_{ij}| = \|\Delta\|_\infty.$$

For the second inequality, by Cauchy–Schwarz on each row $i$,

$$\sum_{j=1}^N |\Delta_{ij}| \le \sqrt{N}\left(\sum_{j=1}^N \Delta_{ij}^2\right)^{1/2} = \sqrt{N}\,\|\Delta_{i,.}\|_2.$$

Moreover,

$$\max_i \|\Delta_{i,.}\|_2 = \max_i \|\Delta^\top e_i\|_2 \le \|\Delta^\top\|_2 \|e_i\|_2 = \|\Delta\|_2.$$

Taking the maximum over $i$ yields

$$\|\Delta\|_\infty = \max_i \sum_j |\Delta_{ij}| \le \sqrt{N} \max_i \|\Delta_{i,.}\|_2 \le \sqrt{N}\,\|\Delta\|_2.$$

**(2) Inverse diagonal control.** Because $\widehat{D} = D + E$ is diagonal, its smallest diagonal entry satisfies

$$\min_i \widehat{D}_{ii} = \min_i(D_{ii} + E_{ii}) \ge \min_i D_{ii} - \max_i |E_{ii}| = s_{\min} - \|E\|_2.$$

If $\|E\|_2 \le s_{\min}/2$, then $\min_i \widehat{D}_{ii} \ge s_{\min}/2 > 0$, so $\widehat{D}$ is invertible and

$$\|\widehat{D}^{-1}\|_2 = \max_i \frac{1}{\widehat{D}_{ii}} \le \frac{1}{s_{\min} - \|E\|_2} \le \frac{1}{s_{\min} - s_{\min}/2} = \frac{2}{s_{\min}}.$$

$\qquad\qquad\square$

Assumption (A1) ensures $D$ has diagonals of order $N$, but we must still control $E$. The next lemma shows that the condition $\|E\|_2 \le s_{\min}/2$ holds with high probability once $L$ and $\beta$ are moderately large.

**Lemma 10** (Concentration bound for $E$). *Recall $E = \widehat{D} - D = \mathrm{diag}((\widehat{S} - S)\mathbf{1})$ and assume (A1): $s_{\min} \ge C_1 N$. Then with probability at least $1 - \delta$,*

$$\|E\|_2 \;\le\; N\sqrt{\tfrac{1}{2L}\log\tfrac{2N^2}{\delta}} \;+\; \frac{4}{\sqrt{2\pi}}\frac{NP}{\beta} \;+\; \mathcal{O}(NPe^{-c\beta}), \qquad c = 2\tanh(1).$$

*In particular, $\|E\|_2 \le \tfrac{1}{2}s_{\min}$ holds with probability at least $1 - \delta$ whenever*

$$L \;\ge\; \frac{8}{C_1^2}\log\frac{2N^2}{\delta}, \qquad \beta \;\ge\; \frac{32}{C_1\sqrt{2\pi}}P,$$

*in which case the exponentially small term $NPe^{-c\beta}$ can be absorbed into constants.*

*Proof.* Since

$$E = \widehat{D} - D = \mathrm{diag}((\widehat{S} - S)\mathbf{1}),$$

we have

$$\|E\|_2 = \max_i \Big| \sum_{j=1}^{N} (\widehat{S}_{ij} - S_{ij}) \Big|.$$

Decompose the kernel error as

$$\widehat{S} - S = (\widehat{S} - \mathbb{E}[\widehat{S}]) + (\mathbb{E}[\widehat{S}] - S) = (\widehat{S} - \mathbb{E}[\widehat{S}]) + \tilde{B},$$

where $\tilde{B}$ is the bias matrix.

Hence for any row $i$,

$$\Big| \sum_j (\widehat{S}_{ij} - S_{ij}) \Big| \le \sum_j |\widehat{S}_{ij} - \mathbb{E}[\widehat{S}_{ij}]| + \sum_j |\tilde{B}_{ij}|$$

$$\le N \max_j |\widehat{S}_{ij} - \mathbb{E}[\widehat{S}_{ij}]| + N \max_j |\tilde{B}_{ij}|.$$

Taking the maximum over $i$ gives

$$\|E\|_2 \le N \max_{i,j} |\widehat{S}_{ij} - \mathbb{E}[\widehat{S}_{ij}]| + N \max_{i,j} |\tilde{B}_{ij}|.$$

Each $\widehat{S}_{ij}$ is an average of $L$ i.i.d. random variables $\widehat{S}_{ij}^{(\ell)} \in [0, 1]$. By Hoeffding's inequality,

$$\Pr(|\widehat{S}_{ij} - \mathbb{E}[\widehat{S}_{ij}]| > \epsilon) \le 2e^{-2L\epsilon^2}.$$

A union bound over all $N^2$ entries yields that with probability at least $1 - \delta$,

$$\max_{i,j} |\widehat{S}_{ij} - \mathbb{E}[\widehat{S}_{ij}]| \le \sqrt{\tfrac{1}{2L}\log\tfrac{2N^2}{\delta}}.$$

Eq. 6 gives the entrywise bias bound

$$\max_{i,j} |\tilde{B}_{ij}| \le \frac{4}{\sqrt{2\pi}}\frac{P}{\beta} + \mathcal{O}(Pe^{-c\beta}).$$

Combining the bounds proves

$$\|E\|_2 \le N\sqrt{\tfrac{1}{2L}\log\tfrac{2N^2}{\delta}} + \frac{4}{\sqrt{2\pi}}\frac{NP}{\beta} + \mathcal{O}(NPe^{-c\beta}).$$

Under assumption (A1), $s_{\min} \ge C_1 N$. Thus $\|E\|_2 \le \tfrac{1}{2}s_{\min}$ is ensured whenever

$$\sqrt{\tfrac{1}{2L}\log\tfrac{2N^2}{\delta}} + \frac{4}{\sqrt{2\pi}}\frac{P}{\beta} \le \frac{C_1}{2}.$$

A sufficient explicit choice is

$$L \geq \frac{8}{C_1^2} \log \frac{2N^2}{\delta}, \qquad \beta \geq \frac{16}{C_1\sqrt{2\pi}}P,$$

while the exponentially small term is absorbed into constants for moderate $\beta$. $\qquad\square$

Now, with row-sums controlled, we relate $\widehat{A}$ and $A$ exactly through a decomposition that isolates the contributions of $\Delta$ in both the numerator and denominator.

**Lemma 11** (Exact perturbation identity and bound).

$$\widehat{A} - A = \widehat{D}^{-1}\Delta + (\widehat{D}^{-1} - D^{-1})S.$$

*Moreover, whenever $\|E\|_2 < s_{\min}$,*

$$\|\widehat{A} - A\|_2 \leq \frac{\|\Delta\|_2}{s_{\min} - \|E\|_2} + \frac{\|S\|_2 \|E\|_2}{s_{\min}(s_{\min} - \|E\|_2)}.$$

*Proof.* Using $\widehat{S} = S + \Delta$ and $\widehat{D} = D + E$,

$$\widehat{A} - A = \widehat{D}^{-1}\widehat{S} - D^{-1}S = \widehat{D}^{-1}\Delta + (\widehat{D}^{-1} - D^{-1})S.$$

For the bound, we apply the submultiplicativity property of norms. When $\|E\|_2 < s_{\min}$, we have $\|D^{-1}\|_2 = 1/s_{\min}$ and $\|\widehat{D}^{-1}\|_2 \leq 1/(s_{\min} - \|E\|_2)$. Moreover,

$$\|\widehat{D}^{-1} - D^{-1}\|_2 = \|\widehat{D}^{-1}(D - \widehat{D})D^{-1}\|_2 \leq \|\widehat{D}^{-1}\|_2 \|E\|_2 \|D^{-1}\|_2 \leq \frac{\|E\|_2}{s_{\min}(s_{\min} - \|E\|_2)}.$$

Hence,

$$\|\widehat{A} - A\|_2 \leq \|\widehat{D}^{-1}\|_2 \|\Delta\|_2 + \|\widehat{D}^{-1} - D^{-1}\|_2 \|S\|_2 \leq \frac{\|\Delta\|_2}{s_{\min} - \|E\|_2} + \frac{\|S\|_2 \|E\|_2}{s_{\min}(s_{\min} - \|E\|_2)}.$$

$\qquad\square$

Specializing Lemma 11 to the regime $\|E\|_2 \leq s_{\min}/2$ (guaranteed w.h.p. by Lemma 10), we obtain a concise spectral bound for $\|\widehat{A} - A\|_2$ in terms of $\|\Delta\|_2$.

**Lemma 12** (Attention deviation). *If $\|E\|_2 \leq s_{\min}/2$, then*

$$\|\widehat{A} - A\|_2 \leq \frac{2\|\Delta\|_2}{s_{\min}} + \frac{2\|S\|_2}{s_{\min}^2}\sqrt{N}\,\|\Delta\|_2.$$

*Proof.* From Lemma 11,

$$\|\widehat{A} - A\|_2 \leq \frac{\|\Delta\|_2}{s_{\min} - \|E\|_2} + \frac{\|S\|_2 \|E\|_2}{s_{\min}(s_{\min} - \|E\|_2)}.$$

Since $\|E\|_2 \leq s_{\min}/2$, it follows that

$$\frac{1}{s_{\min} - \|E\|_2} \leq \frac{1}{s_{\min}/2} = \frac{2}{s_{\min}}.$$

Substituting this bound gives

$$\|\widehat{A} - A\|_2 \leq \frac{2\|\Delta\|_2}{s_{\min}} + \frac{2\|S\|_2}{s_{\min}^2}\|E\|_2$$

By Lemma 9(1), $\|E\|_2 \leq \|\Delta\|_\infty \leq \sqrt{N}\,\|\Delta\|_2$. Therefore,

$$\|\widehat{A} - A\|_2 \leq \frac{2\|\Delta\|_2}{s_{\min}} + \frac{2\|S\|_2}{s_{\min}^2}\sqrt{N}\,\|\Delta\|_2,$$

which proves the claim. $\qquad\square$

Finally, we translate attention deviation into end-to-end output deviation by a single multiplication with the value matrix $V$, yielding the main finite-sample guarantee.

**Theorem 13** (End-to-end output error). *Let $V \in \mathbb{R}^{N \times d}$ be the value matrix. With probability at least $1 - \delta$, if $\|E\|_2 \leq s_{\min}/2$ then*

$$\|\widehat{O} - O\|_F \leq \left(\frac{2}{s_{\min}} + \frac{2\|S\|_2\sqrt{N}}{s_{\min}^2}\right)\left(\frac{4}{\sqrt{2\pi}}\frac{NP}{\beta} + \mathcal{O}(NPe^{-c\beta}) + 4\frac{N}{\sqrt{L}}\sqrt{\log\frac{2N}{\delta}} + \frac{4}{3}\frac{N}{L}\log\frac{2N}{\delta}\right)\|V\|_F,$$

*where $c = 2\tanh(1)$.*

*Proof.* By the estimator identity, $\widehat{O} = \widehat{A}V$ and $O = AV$, hence using submultiplicativity of the Frobenius norm,

$$\|\widehat{O} - O\|_F = \|(\widehat{A} - A)V\|_F \leq \|\widehat{A} - A\|_2\,\|V\|_F.$$

Under the condition $\|E\|_2 \leq s_{\min}/2$, Lemma 12 (Attention deviation) gives

$$\|\widehat{A} - A\|_2 \leq \left(\frac{2}{s_{\min}} + \frac{2\|S\|_2\sqrt{N}}{s_{\min}^2}\right)\|\widehat{S} - S\|_2.$$

Applying Theorem 6 (Kernel deviation) yields, with probability at least $1 - \delta$,

$$\|\widehat{S} - S\|_2 \leq \|\tilde{B}\|_2 + 4\frac{N}{\sqrt{L}}\sqrt{\log\frac{2N}{\delta}} + \frac{4}{3}\frac{N}{L}\log\frac{2N}{\delta}.$$

Finally, substitute the explicit bias bound from Lemma 8:

$$\|\tilde{B}\|_2 \leq \frac{4}{\sqrt{2\pi}}\frac{NP}{\beta} + \mathcal{O}(NPe^{-c\beta}), \qquad c = 2\tanh(1).$$

Combining the three equations proves the stated inequality:

$$\|\widehat{O} - O\|_F \leq \left(\frac{2}{s_{\min}} + \frac{2\|S\|_2\sqrt{N}}{s_{\min}^2}\right)\left(\frac{4}{\sqrt{2\pi}}\frac{NP}{\beta} + \mathcal{O}(NPe^{-c\beta}) + 4\frac{N}{\sqrt{L}}\sqrt{\log\frac{2N}{\delta}} + \frac{4}{3}\frac{N}{L}\log\frac{2N}{\delta}\right)\|V\|_F. \tag{7}$$

$\square$

*Proof of Theorem 2.* Under assumptions (A1) and (A2), we have $s_{\min} \geq C_1 N$ and $\|S\|_2 \leq C_2 N$. Therefore, the prefactor on the r.h.s. of Eq. equation 7 boils down to

$$\frac{2}{s_{\min}} + \frac{2\|S\|_2\sqrt{N}}{s_{\min}^2} \leq \frac{2}{C_1 N} + \frac{2C_2 N\sqrt{N}}{C_1^2 N^2} = \mathcal{O}\left(\frac{1}{\sqrt{N}}\right). \tag{8}$$

Therefore, combining eqs. equation 7 and equation 8 yields

$$\begin{aligned}
\|\widehat{O} - O\|_F &= \mathcal{O}\left(\frac{1}{\sqrt{N}}\right)\left(\frac{NP}{\beta} + \frac{N}{\sqrt{L}}\sqrt{\log\frac{2N}{\delta}} + \frac{N}{L}\log\frac{2N}{\delta} + NPe^{-c\beta}\right)\|V\|_F \\
&= \mathcal{O}\left(\frac{P\sqrt{N}}{\beta} + \sqrt{\frac{N}{L}\log\frac{2N}{\delta}} + \frac{\sqrt{N}}{L}\log\frac{2N}{\delta} + P\sqrt{N}\,e^{-c\beta}\right)\|V\|_F. \tag{9}
\end{aligned}$$

Dividing both sides of Eq. equation 9 by $\sqrt{N}$ to express the bound in terms of the per-token RMS error gives

$$\|\widehat{O} - O\|_{\mathrm{rms}} \leq \mathcal{O}\left(\frac{P}{\beta} + \sqrt{\frac{\log(2N/\delta)}{L}} + \frac{1}{L}\log\frac{2N}{\delta} + Pe^{-c\beta}\right)\|V\|_F.$$

To compare the last two variance terms, observe that

$$\frac{\frac{1}{L}\log\frac{2N}{\delta}}{\sqrt{\frac{\log(2N/\delta)}{L}}} = \sqrt{\frac{\log(2N/\delta)}{L}}.$$

Hence, whenever $L \geq \log(2N/\delta) = \Theta(\log N)$, the $(1/L)\log(2N/\delta)$ term is asymptotically dominated by $\sqrt{\log(2N/\delta)/L}$ and can therefore be absorbed into it. The exponentially small correction $Pe^{-c\beta}$ is also negligible for moderate values of $\beta$. Absorbing all constants and the mild difference between $\log(2N/\delta)$ and $\log(N/\delta)$ into the Big-$\mathcal{O}$, we obtain

$$\|\widehat{O} - O\|_{\mathrm{rms}} = \mathcal{O}\left(\frac{P}{\beta} + \sqrt{\frac{\log(N/\delta)}{L}}\right)\|V\|_F,$$

with probability at least $1 - \delta$. This completes the proof. $\square$

## B.4 CAUSAL RACE ATTENTION

---

**Algorithm 2** RACE Attention (causal)

---

**Input:** $Q, K, V \in \mathbb{R}^{N \times d}$; number of hash tables $L$; number of hyperplanes $P$; temperature $\beta > 0$.
**Output:** $\widehat{O} \in \mathbb{R}^{N \times d}$.

1: **for** $\ell = 1, \dots, L$ **do**
2:      Draw $W^{(\ell)} \in \mathbb{R}^{P \times d}$          // $P$ random hyperplanes
3:      Define the corner set $\mathcal{V} = \{\pm 1\}^P$ $(R = 2^P)$ with $v_r \in \{\pm 1\}^P$      // $R$ corners
4:      Build $\Phi_Q^{(\ell)}, \Phi_K^{(\ell)} \in \mathbb{R}^{N \times R}$ with rows

$$[\phi^{(\ell)}(x)]_r = \frac{\exp\{\beta(\tanh(W^{(\ell)}x))^\top v_r\}}{\sum_{r'} \exp\{\beta(\tanh(W^{(\ell)}x))^\top v_{r'}\}}, \quad x \in \{Q_i, K_j\}.$$

5:      Initialize cumulative bucket statistics: $A_{\text{cum}}^{(\ell)} \leftarrow \mathbf{0}_R \in \mathbb{R}^R$, $B_{\text{cum}}^{(\ell)} \leftarrow \mathbf{0}_{R \times d} \in \mathbb{R}^{R \times d}$.
6:      **for** $t = 1, \dots, N$ **do**
7:          $\Phi_K^{(\ell)}[t, :] \in \mathbb{R}^R, \quad V_t \in \mathbb{R}^d$
8:          $A_{\text{cum}}^{(\ell)} \leftarrow A_{\text{cum}}^{(\ell)} + (\Phi_K^{(\ell)}[t, :])^\top$          // $\mathbb{R}^R$
9:          $B_{\text{cum}}^{(\ell)} \leftarrow B_{\text{cum}}^{(\ell)} + (\Phi_K^{(\ell)}[t, :])^\top V_t$          // $\mathbb{R}^{R \times d}$
10:         $\Phi_Q^{(\ell)}[t, :] \in \mathbb{R}^R$
11:         $\text{num}_t^{(\ell)} \leftarrow \Phi_Q^{(\ell)}[t, :] B_{\text{cum}}^{(\ell)}$          // $(1 \times R) \cdot (R \times d) = \mathbb{R}^d$
12:         $\text{den}_t^{(\ell)} \leftarrow \Phi_Q^{(\ell)}[t, :] A_{\text{cum}}^{(\ell)}$          // $(1 \times R) \cdot (R) = \mathbb{R}$
13:      **end for**
14: **end for**
15: For each $t = 1, \dots, N$:

$$\text{Num}_t = \frac{1}{L} \sum_{\ell=1}^L \text{num}_t^{(\ell)} \in \mathbb{R}^d, \qquad \text{Den}_t = \frac{1}{L} \sum_{\ell=1}^L \text{den}_t^{(\ell)} \in \mathbb{R}, \qquad \widehat{O}_t = \frac{\text{Num}_t}{\text{Den}_t} \in \mathbb{R}^d.$$

16: Return: $\widehat{O} = \begin{bmatrix} \widehat{O}_1^\top \\ \vdots \\ \widehat{O}_N^\top \end{bmatrix} \in \mathbb{R}^{N \times d}.$

---

We implemented the causal version (see Algorithm 2) efficiently using OpenMP/CUDA-based parallelization rather than a naive nested-loop approach. Each hash table is processed in a separate thread with its own cumulative bucket arrays, and updates are performed incrementally in a single left-to-right scan. This avoids redundant recomputation at every step using torch.cumsum() and enables CPU/GPU-level parallel execution with negligible synchronization overhead.

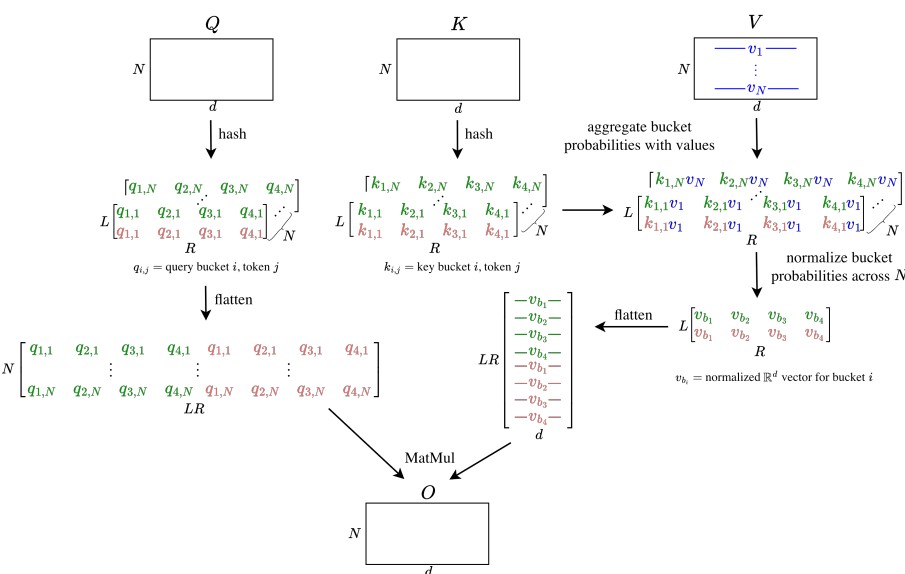

Figure 7: RACE Attention pipeline from the inputs $Q, K, V \in R^{N \times d}$ to the output $O \in R^{N \times d}$: queries/keys are soft-hashed into $R$ buckets across $L$ tables, keys/values form per-bucket summaries, and each query mixes the bucket summaries to produce $O$.

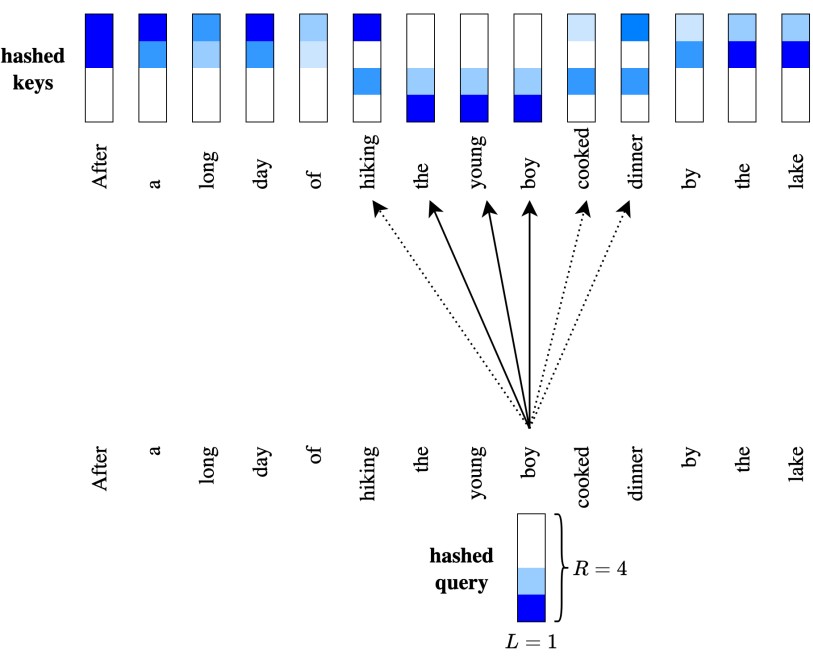

Figure 8: An intuitive schematic of how RACE Attention runs with $L$ hash tables and $R$ buckets per table. Similarity between Queries and Keys is highest if they both assign a higher mass to same buckets across all hash tables.

