# OpenReview forum: "RACE Attention: A Strictly Linear-Time Attention Layer for Training on Outrageously Large Contexts"
_ICLR.cc/2026/Conference — ICLR 2026 Poster_

### Official Review · Reviewer_if3U · 2025-10-31

**Soundness:** 3
**Presentation:** 2
**Contribution:** 3
**Rating:** 6
**Confidence:** 4

**Summary:**

This paper describes RACE attention as a linear-time alternative to softmax attention for very long contexts. The main idea is to replace softmax with powers of angular similarity, and then approximate this term using RACE sketches. To do this, the algorithm uses soft LSH so that its differentiable. This achieves far reduced complexity versus quadratic for standard attention, as is common in most methods for self-attention approximation. What is nice is that the experiments are broad and cover language modeling, masked LM, and classification. In this context, scaling experiments show processing of tens of millions of tokens on CPU and GPU for a single attention layer's forward-backward pass. This will be the main highlight of this work for most readers.

**Strengths:**

1. The scaling experiments are quite impressive. Regardless of my other comments below, this is a good practical contribution. Also, it is interesting that CPU-based RACE is viable and in some regimes can do better than FlashAttention. This point about algorithmic efficiency versus hardware acceleration could really be a main message of the paper (more on this below). In any case, reaching 50M/75M tokens is definitely a strength (but in the current version of the paper, this comes with some disclaimer).

2. The experimental breadth is very good. Both CPU and GPU kernels with OpenMP are mentioned. This is a strong engineering effort and if code is provided, it can benefit many groups working in this area.

3. Experimental verification of how increasing degree can mimic exponential behavior in this setting is useful. Some analysis is included for the bias-variance to guide the choices in the sketching component. This is all good.

**Weaknesses:**

1. I am a bit confused by the numerous instances of "stress test" and therefore it unclear what the scaling experiments actually show. When stress testing 1 forward-backward pass with the multi-head attention layer, is this timing a single layer, not end-to-end model training? If so, the 75M token claim is for one attention operation, not training the full model? Is this paper only describing benchmarking the primitive or does any model work at these lengths? The reason for this question is the title "outrageously large context windows" -- is this only for the stress tests? The most reasonable reading of the title suggests full model capability.

2. I am having trouble understanding the tables on page 8. Is angular expected to be better than RACE?

3. The paper https://proceedings.mlr.press/v139/zeng21a.html uses related ideas and also seems motivated by similar upstream papers. Another one is https://aclanthology.org/2022.iwslt-1.4.pdf. The positioning of this work on page 4/5 should at least describe how they differ.

**Questions:**

1. Minor: Is adapting the analysis to causal masking relatively easy (but hasn't been worked out yet) or does one run into problems?
2. check some of the references above. There may be others.

---

> ### Author Response · Authors · 2025-11-20
> **Response (1/2) to Reviewer if3U**
>
> > Q1. "I am a bit confused by the numerous instances of "stress test" and therefore it unclear... The most reasonable reading of the title suggests full model capability."
>
> A1. We thank the reviewer for raising this important point. To clarify unambiguously, our scaling experiments stress-test only the attention mechanism itself, not the end-to-end training of a full Transformer model. This experimental design follows well-established practice in prior work on scalable attention mechanisms, including HyperAttention [1] and Performer [2].
>
> Our stress tests benchmark a single forward–backward pass of a multi-head attention layer under fixed hyperparameters (the same ones used in our accuracy evaluations). Thus, the reported 75M-token capability refers solely to the maximum sequence length for which the attention primitive remains computationally feasible, not for training a full Transformer model at that length.
>
> We do not train any end-to-end model at 75M tokens. Rather, the purpose of these experiments is to evaluate the scalability of the attention operation in isolation, which is the dominant computational bottleneck in long-context models.
>
> This decomposition is deliberate and directly motivated by the behavior of long-context architectures. At extremely large sequence lengths:
>
> - The attention operation dominates both memory and compute.
>
> - Many existing attention variants fail or become infeasible far earlier.
>
> - Demonstrating that the attention mechanism itself can scale to tens of millions of tokens is a necessary prerequisite before any full model can be trained in this regime.
>
> Lastly, when we refer to “outrageously large context windows,” we are referring specifically to the attainable context range of the attention mechanism under practical forward–backward compute. While end-to-end Transformer training at 75M tokens is not yet standard practice, our results indicate that our proposed RACE Attention mechanism indeed brings such regimes significantly closer to feasibility.
>
> **[1] Insu Han, Rajesh Jayaram, Amin Karbasi, Vahab Mirrokni, David P. Woodruff, Amir Zandieh. HyperAttention: Long-Context Attention in Near-Linear Time. ICLR 2024.**
>
> **[2] Krzysztof Choromanski, Valerii Likhosherstov, David Dohan, Xingyou Song, Andreea Gane, Tamas Sarlos, Peter Hawkins, Jared Davis, Afroz Mohiuddin, Lukasz Kaiser, David Belanger, Lucy Colwell, Adrian Weller. Rethinking Attention with Performers. ICLR 2021.**
>
> > Q2. Clarifications for Tables on page 8.
>
> A2. Yes, Angular is expected to be slightly better than RACE in general, because RACE approximates the sharpened angular kernel.
>
> > Q3. Comparison with YOSO [3] and LSH Attention [4].
>
> A3. We thank the reviewer for highlighting these related works. Both papers are indeed relevant, and we appreciate the opportunity to clarify our positioning relative to them. We discuss and outline the key differences between RACE and YOSO Attention in Section 1 (lines 53–61) and further elaborate in our response to Reviewer W9FA (Q1). Lastly, we do note that the LSH Attention paper [4] naturally fits within structural-sparsity approaches, and we now explicitly cite it in the "Sparsity is Complementary" subsection of Section 1. This work applies Reformer-style LSH attention to cross-attention within neural machine translation models. Its goal is to leverage sparsity patterns to reduce attention cost in encoder-decoder architectures, and is therefore complementary to ours. Our approach focuses on improving the efficiency of the core dense self-attention mechanism itself in a mathematically principled way. For this reason we view such sparsity-based methods, including [4], as orthogonal to our method.
>
> **[3] Zhanpeng Zeng, Yunyang Xiong, Sathya N. Ravi, Shailesh Acharya, Glenn Fung, Vikas Singh. You Only Sample (Almost) Once: Linear Cost Self-Attention Via Bernoulli Sampling. ICML 2021.**
>
> **[4] Frithjof Petrick, Jan Rosendahl, Christian Herold, Hermann Ney. Locality-sensitive hashing for long context neural machine translation. IWSLT 2022.**

---

> > ### Comment · Reviewer_if3U · 2025-11-26
> > **framing, positioning and experiments**
> >
> > Thanks for your clarification. Let me comment briefly on a few points, which after reading your response, have reduced my support for this paper.
> >
> > I am quite concerned by two framing issues.
> >
> > First, your title in my opinion misrepresents what is really demonstrated. A context window to me refers to what a complete model can process end-to-end. This does not mean an isolated primitive. You have stress-tested a single attention layer at 75M tokens. This is impressive but at the same time, very different from training a model at that scale where memory compounds across layers. In your response, you cite HyperAttention/Performer as precedent for primitive benchmarking. I agree. But neither of these papers makes such an aggressive claim by using an attention grabbing title. Your defense is that this is a necessary prerequisite. Again I agree. But showing a prerequisite does not line up with your title. The framing oversells the practical capability and what systems can actually be built today.
> >
> > I am also disappointed with the somewhat superficial engagement with the YOSO work in your response. I think that the novelty is oversold in your paper. Does YOSO also not use a similar kernel as its target and discuss unbiased estimation?  Your work is adding concentration bounds via standard proof machinery. Essentially more tables and higher temperature is better. In my reading, this paper is making the discrete bucketing differentiable via soft assignments. This is good but the positioning of claiming algorithmic novelty while dismissing prior work even though it was pointed out in two separate reviews is confusing.
> >
> > The systems-level work is impressive and represents the main strength. But this would need, at a minimum, an honest framing that this paper makes LSH-attention already presented in earlier works practical and highlighting the main contribution by situating it correctly with the prior works, rather than overselling theory. This would be a very different paper.

---

> ### Author Response · Authors · 2025-11-20
> **Response (2/2) to Reviewer if3U**
>
> > Q4. Challenges in extending the current analysis to causal settings.
>
> A4. Thank you for raising this important question. Algorithmically, adapting RACE Attention to causal masking is straightforward, and we already employ a prefix-scan (causal) version in all autoregressive experiments (see Algorithm 2 in the Appendix and Tables 6, 12). The modification concerns only how bucket summaries are accumulated.
>
> In the non-causal setting (Algorithm 1), each hash table $\ell$ uses global bucket summaries
> $$
> A^{(\ell)} = \sum_{j=1}^N \phi^{(\ell)}(K_j),
> \qquad
> B^{(\ell)} = \sum_{j=1}^N \phi^{(\ell)}(K_j)V_j,
> $$
> which are shared across all query positions.
>
> However,  under causal masking, these become prefix-cumulative quantities,
> $$
> A_{\mathrm{cum}}^{(\ell)}(t) = \sum_{j \le t} \phi^{(\ell)}(K_j),
> \qquad
> B_{\mathrm{cum}}^{(\ell)}(t) = \sum_{j \le t} \phi^{(\ell)}(K_j)V_j,
> $$
> and the attention estimate at time $t$ uses
> $$
> \text{Numerator: }
> \phi(Q_t)^\top B_{\mathrm{cum}}^{(\ell)}(t),
> \qquad
> \text{Denominator: }
> \phi(Q_t)^\top A_{\mathrm{cum}}^{(\ell)}(t).
> $$
> The feature maps $\phi(Q_i), \phi(K_j)$, the Soft-RACE hashing, and the kernel estimator all remain unchanged; only the accumulation rule differs. Thus, the causal variant is a light modification of the non-causal algorithm. The difficulty lies in the theory rather than in the implementation. In the non-causal case, all queries share the same normalization mass $\phi(Q_i)^\top A^{(\ell)}$, enabling a uniform variance analysis. In contrast, under causal masking, the normalization term $\phi(Q_t)^\top A_{\mathrm{cum}}^{(\ell)}(t)$ grows with $t$, and both numerator and denominator depend on the same evolving cumulative buckets. This introduces position-dependent variance and temporal dependencies that are absent in the non-causal setting, preventing a direct extension of the proof of Theorem 2. Therefore, a rigorous analysis of causal RACE Attention is an important direction for future work.

---

> ### Author Response · Authors · 2025-11-27
> **Response to Reviewer if3U on Framing, Positioning and Experiments**
>
> > Q5. Discussion on Framing, Positioning and Experiments.
>
> We sincerely thank the reviewer for clearly articulating the mismatch between our original title and the actual scope of our contributions. Our intention was to highlight both the aspects: Time and Accuracy in long-context training and Scalability at the primitive level. We now fully understand that the term “context windows” in the title can indeed be misleading, as it may suggest full end-to-end model capability at tens of millions of tokens, whereas our scaling experiments stress-test only the attention mechanism itself at such extreme lengths. We appreciate the reviewer’s guidance in identifying this framing issue. In response, and to better align the title with the precise scope of our work, we have revised the title to: **"RACE Attention: A Linear-Time Attention Mechanism for Long-Sequence Training with Extreme-Length Attention-Layer Scaling"**
>
> This revised title avoids implying full-model training at extreme context lengths, while still capturing the two key aspects we intended to highlight:
> - The ability of our method to train effectively on long sequences in strictly linear time, and
> - The ability of the underlying attention mechanism to scale to extreme sequence lengths.
>
> We believe that framing it this way more accurately reflects the balance of algorithmic, theoretical, and systems-level contributions in the paper.
>
> > Q6. Detailed comparison with YOSO.
>
> We appreciate the reviewer’s emphasis on the need for a more thorough discussion of YOSO. We recognize that our earlier response did not clearly convey the relationship between the two methods. The reviewer is correct that both YOSO and our approach operate on the same LSH-induced powered angular kernel and that YOSO discusses unbiased estimation of this kernel. We fully acknowledge this connection and have revised the paper (lines 51-59 and 263-272) to make it more explicit.
>
> At the same time, while the underlying kernel family is shared, the two approaches differ significantly in what is being estimated, how the estimation is performed, and the resulting properties of the attention output. Standard attention's formula (Eq. 3 in our paper) can be written as:
> $$
> O_i=\frac{\sum_{j=1}^N sim\left(Q_i, K_j\right) V_j}{\sum_{j=1}^N sim \left(Q_i, K_j\right)}
> $$
> and for both YOSO and RACE, the similarity is given by the LSH-induced powered angular kernel,
> $$
> \mathrm{sim}(Q_i, K_j) = \left(1 - \frac{\cos^{-1}\left(\frac{Q_i^\top K_j}{\lVert Q_i\rVert\, \lVert K_j\rVert}\right)}{\pi}\right)^\gamma.
> $$
> YOSO constructs a Bernoulli collision matrix $B$ via hard LSH hashing, where $(BV)\_i$ is an unbiased estimator of the _unnormalized_ kernel numerator $\sum_{j=1}^N sim(Q_i, K_j)V_j$. However, YOSO does not estimate the denominator $\sum_{j=1}^N \mathrm{sim}(Q_i, K_j)$ and their final output is obtained through post-hoc $\ell_2$ normalization, which does not correspond to the traditional normalization in attention. Furthermore, since the hard LSH is non-differentiable, YOSO cannot backpropagate through the hashing operation. As noted in Section 3.3 of their paper, the true derivative of the collision probability is numerically unstable and diverges as the similarity score $Q_i^TK_j$ approaches $1$, so YOSO replaces it with a _surrogate lower-bound gradient_ that is estimated using additional Bernoulli hash samples. Thus, YOSO's estimator is not differentiable in the usual sense and relies on surrogate-gradient approximations for training, incurring a quadratic time in embedding dimension $d$.
>
> In contrast, our work explicitly utilizes the above formulation which is a normalized kernel attention and uses RACE sketches to approximate _both_ the numerator and denominator in strictly linear time. Our novel soft-bucketization strategy provides a smooth and fully differentiable relaxation of hyperplane hashing, enabling stable end-to-end training without surrogate gradients. Averaging across $ \mathrm{~L} $ independent RACE tables yields a low-variance estimator, and Theorem $2$ provides an explicit bias-variance guarantee for the resulting attention output. Finally, unlike YOSO, we show strong accuracy with bidirectional attention up to 64K sequence lengths, and our construction naturally supports a causal variant on which we show strong performance on the standard WikiText-103 benchmark.
>
> In the revised version, we now explicitly describe how YOSO serves as an important precursor within the same kernel family and position our contribution as introducing a differentiable, provably accurate RACE-based estimator for normalized attention with long-sequence trainability and extreme-length scalability. We thank the reviewer for prompting us to clarify this relationship and hope this addresses the concern.

---

### Official Review · Reviewer_eQBU · 2025-11-01

**Soundness:** 3
**Presentation:** 2
**Contribution:** 2
**Rating:** 4
**Confidence:** 3

**Summary:**

This paper introduces RACE Attention, a method to address the quadratic time and memory complexity of standard softmax attention. The authors propose replacing the exponential softmax kernel with a high-degree monomial of an angular (cosine) similarity kernel. This specific kernel choice allows them to leverage Locality Sensitive Hashing (LSH) and Repeated Arrays-of-Count Estimators (RACE) sketches to compute the attention output in linear time and space complexity.

**Strengths:**

1.  The primary contribution and strength of this paper are the scaling results. Figure 5 shows that RACE on a CPU can outperform FlashAttention on a high-end GPU at massive sequence lengths, is a compelling demonstration of the algorithm's effect over hardware acceleration.

2. The paper is well-written and easy to follow.

3. The theoretical result also provides a nice bias-variance trade-off of their approach.

**Weaknesses:**

1. The paper seems to be lacking some important baselines. The authors compare their result to FlashAttention, however, at the moment FlashAttn 2 and 3 are also available that performs much faster and are not included in the comparison. Moreover, the paper focuses on alternatives to softmax and is for example lacking a comparison to Sigmoid Attention which also provides a simple kernel implementation.

2. The paper is a bit vague and ambiguous in their main algorithm. The authors argue that they use cosine kernel to prevent the exponential of softmax and be able to use RACE sketch. However, it seems that Algorithm 1 is still trying to implement softmax. Am I misunderstanding this? Technically, it seems that the connection between the features $\phi$ and the angular attention is never clearly made.

**Questions:**

1. Can authors elaborate on how to choose $\gamma$? Would it be through a hyperparameter search or is there a principled way of approximating a good value for it?

2. Once more question on $\gamma$, could authors provide any sensitivity analysis of how the final result changes with respect to the small changes in $\gamma$? Perhaps another useful figure would be to use the data from Fig 2 and plot the distribution of the attention distances between softmax and the angular attention to see how it varies as $\gamma$ is changed.

---

> ### Author Response · Authors · 2025-11-20
> **Response (1/2) to Reviewer eQBU**
>
> > Q1. Comparison with additional baselines.
>
> A1. Thank you for raising this important point. We appreciate the reviewer for highlighting the ambiguity in our FlashAttention baselines. First, we clarify that all our experiments use **FlashAttention-2** as the GPU baseline. In the revised paper, we additionally include results for **FlashAttention-3** and **Sigmoid Attention** in the scaling comparisons (figs. 5–6). For the accuracy evaluations (tables 1, 2, 4), we report Sigmoid Attention, and omit FlashAttention-3 since it achieves the same accuracy as FlashAttention-2. We also report how our RACE Attention achieves significant speedups on CPU and GPU over these baselines in figs. 7 and 8 in the Appendix, following the evaluation style of [1] (see fig. 4 in [1]).
>
> Although FlashAttention-2/3 and Sigmoid Attention are highly optimized kernels, they still scale quadratically with sequence length and thus become impractical in the long-context regime. We omit training Sigmoid Attention at extremely long lengths (8K-64K), as it is quadratic regardless of implementation, and FlashAttention-2 already serves as a representative quadratic baseline. In contrast, RACE Attention continues to scale efficiently far beyond the maximum sequence lengths reachable by these methods.
>
> **[1] Insu Han, Rajesh Jayaram, Amin Karbasi, Vahab Mirrokni, David P. Woodruff, Amir Zandieh. HyperAttention: Long-Context Attention in Near-Linear Time. ICLR 2024.**
>
> > Q2. The use of Softmax in Algorithm 1.
>
> A2. Thank you for raising this important clarification. Our use of softmax in Algorithm 1 (see step 4) is not the softmax attention mechanism used in standard Transformers. In our Soft RACE, the softmax operation appears only as a local smoothing machinery to make the LSH bucket assignments differentiable. It replaces the hard {-1,1} indicator of a bucket with a continuous distribution over the $R=2^P$ corners, capturing how a token’s mass is shared among the $R$ buckets. Therefore, we emphasize that the softmax step in Algorithm 1 has nothing to do with the exponential weighting of queries and keys in softmax attention. The actual similarity function we approximate is the powered angular kernel. The connection to RACE arises because classical RACE sketches ([2], [3]) estimate powers of LSH collision kernels, and Soft RACE preserves this structure while making the assignments differentiable through a `softmax+tanh` operation. Thus, the "softmax" in Algorithm 1 serves only to provide soft bucketization, not softmax attention per se. We appreciate the opportunity to clarify this distinction and have added a paragraph explaining this concept on lines 290-298.
>
> **[2] Benjamin Coleman, Anshumali Shrivastava. Sub-linear RACE Sketches for Approximate Kernel Density Estimation on Streaming Data. WWW 2020.**
>
> **[3] Benjamin Coleman, Richard G Baraniuk, Anshumali Shrivastava. Sub-linear Memory Sketches for Near Neighbor Search on Streaming Data. ICML 2020.**

---

> ### Author Response · Authors · 2025-11-20
> **Response (2/2) to Reviewer eQBU**
>
> > Q3. The connection between feature maps and Angular Attention.
>
> Thank you for this insightful question. Here we clarify how our soft feature map $\phi$ connects to the $P$-powered angular kernel. A classical RACE sketch draws $P$ random hyperplanes in $\mathbb{R}^d$. Each hyperplane has two sides, so together they create a sign pattern in {$\\pm 1$}$^P$ that describes on which side of each hyperplane a point lies. Stacking the hyperplanes into $ W \in \mathbb{R}^{P \times d}$ and for any $x\in\mathbb{R}^{d}$, the simple "hard" feature map is defined by
> $ \phi_{\text {hard }}(x):=\operatorname{sign}(W x) \in \\{\pm1\\}^P.
> $
>
> Two points $Q_i$ and $K_j$ share the same feature-map output when they fall on the same side of every hashing hyperplane; for angular LSH (SimHash), this occurs with probability
>
> $$
> \Pr\left[\phi_{\text{hard}}(Q_i)=\phi_{\text{hard}}(K_j)\right]
> = S_{ij}
> := \left(1 - \frac{\cos^{-1}\left(\frac{Q_i^\top K_j}{\lVert Q_i\rVert\,\lVert K_j\rVert}\right)}{\pi}\right)^P,
> $$
>
> which is exactly the $P$-powered angular kernel. This gives the ideal $N \times N$ similarity matrix $S$ with $S_{ij}$ defined above. Soft RACE keeps this geometric structure intact, but avoids making a hard $\pm 1$ assignment. Instead, we compute a soft sign vector $\tanh (W x) \in[-1,1]^P$, which tells us how strongly $x$ lies on each side of each hyperplane. We then compare this soft sign vector to all $2^P$ possible {$\\pm 1$}$^P$ patterns and convert the similarities into a probability distribution (via softmax). This produces the soft feature map $\phi(x) \in \mathbb{R}^{R},$ which spreads weight over the regions that best match the smoothed signs of $x$. Consequently,
> $\phi(Q_i)^{\top} \phi(K_j)$
> acts as a soft analogous to the collision event $\phi_{\text {hard }}(Q_i)=\phi_{\text {hard }}(K_j)$, structurally aligned with the same powered angular kernel. These soft similarities define
>
> $$ \hat{S}_{i j}^{(\ell)}=\phi^{(\ell)}\left(Q_i\right)^{\top} \phi^{(\ell)}\left(K_j\right),$$
>
> which are then averaged across $L$ tables to obtain $\widehat S =\tfrac{1}{L}\sum_{\ell=1}^L \widehat S^{(\ell)}$.
> Theorem 2 then shows how this approximate kernel $\hat{S}$ propagates to the final attention output $\hat{O}$. We included this discussion on lines 296-303 and 312-323.
>
> > Q4. Details on how to choose $\gamma$ in Angular vs. RACE Attention.
>
> A4. We thank the reviewer for this important question. In Angular Attention, the sharpening exponent $\gamma$ is an independent hyperparameter; one typically selects $\gamma$ by inspecting attention heatmaps (e.g., $\gamma = 8$-10 already yields softmax-level sharpness; see updated figs. 2, 3). In RACE Attention, however, $\gamma$ is not a free hyperparameter, instead RACE Attention has $P$ (number of hyperplanes) and $L$ (number of hash tables) as the hyperparameters.
>
> A key contribution of our method is the construction of learnable _Soft RACE_, a smooth, differentiable relaxation of RACE hashing ([1], [2]) that enables a linear-time approximation to a powered angular kernel. In a classical RACE sketch, concatenating $P$ random hyperplanes yields collision probabilities proportional to the $P$-th power of the angular kernel. **Soft RACE preserves this property**, so the effective sharpening exponent emerges directly from the hash construction: $\gamma = P$.
>
> Thus, RACE Attention offers a principled and significantly lower-dimensional design space. In all of our experiments, we restrict $P \in$ {$2,3,4,5$}, which consistently provides strong accuracy and induces an exponential-like sharpening on $[0,1]$, closely matching softmax behavior. Larger $P$ values further sharpen the kernel but are rarely necessary. Meanwhile, the number of hash tables $L$ independently controls variance, as formalized in Theorem 2. Therefore, $\gamma$ in the context of RACE Attention does not require a hyperparameter search. It is fully determined by the choice of $P$.
>
> > Q5. Sensitivity analysis with respect to $\gamma$.
>
> A5. We thank the reviewer for the helpful suggestion. Visualizing the sensitivity of $\gamma$ is indeed valuable, so we now plot the Frobenius error between Softmax and Angular Attention as a function of $\gamma$ in fig. 2 of the updated paper. We can see that the error sharply decreases as $\gamma$ increases, demonstrating that softmax-level sharpness can be achieved with modest polynomial degree (_e.g.,_ $\gamma = 8$).

---

> > ### Comment · Reviewer_eQBU · 2025-11-25
> > **Acknowledging Authors' Rebuttal**
> >
> > Thank you for the detailed response and clarification. Most of my questions are resolved now, I raised my score.

---

> > > ### Author Response · Authors · 2025-11-27
> > > **Thank You Reviewer eQBU!**
> > >
> > > Thank you for the thoughtful questions and for updating your assessment. We appreciate your time and constructive feedback.

---

### Official Review · Reviewer_W9FA · 2025-11-01

**Soundness:** 2
**Presentation:** 3
**Contribution:** 2
**Rating:** 2
**Confidence:** 4

**Summary:**

This paper introduces a novel linear-time attention mechanism. The approach replaces the exponential softmax kernel with a monomial of cosine similarity raised to a power, enabling approximation through randomized projections. By leveraging angular similarity, Locality-Sensitive Hashing, the authors propose an efficient that enables outrageously large context windows  up to 75 million tokens on CPUs and 12 million on GPUs.

**Strengths:**

1. This method enables linear-time and memory-efficient attention that scales to tens of millions of tokens on standard hardware, which is impressive.
2. The algorithm is simple, differentiable, and can serve as a drop-in replacement for softmax attention.

**Weaknesses:**

1. **This paper is very similar to YOSO [1] (for example, the finding the similarity between equation (1) and (2) in the text, the use of LSH in estimating the similarity function, the algorithm of estimating attention outputs via hashtables), but this paper does not discuss and contrast with [1].**
2. The experiments only show model accuracy on short sequence lengths (< 8K). What about longer sequences?
3. The efficiency results in Figure 3 are not very meaningful as any linear attentions can be extremely efficient by tuning their hyperparameters. For example, for $\phi(Q) \phi(K)^T$ type attention, by setting the output dimension of $\phi$ to be 1, its efficiency can beat any other methods. To show efficiency, the runtime and memory results should be coupled with the corresponding accuracy results.
4. Figure 5 has the same issue, what about the accuracy?

**If the authors can address my concerns, I am willing to raise my score.**

[1] Zhanpeng Zeng, Yunyang Xiong, Sathya N. Ravi, Shailesh Acharya, Glenn Fung, Vikas Singh. You Only Sample (Almost) Once: Linear Cost Self-Attention Via Bernoulli Sampling. ICML 2021.

**Questions:**

see weakness section.

---

> ### Author Response · Authors · 2025-11-20
> **Response (1/2) to Reviewer W9FA**
>
> > Q1. YOSO Attention [1] discussion.
>
> A1. We thank the reviewer for highlighting the connection to YOSO. We discuss and outline the key differences between RACE and YOSO in the revised paper (lines 51-59, 263-272). While both approaches employ LSH to approximate attention, the underlying mechanisms and theoretical foundations differ substantially.
>
> **1.) Kernel and Estimator:** YOSO relies on _Bernoulli sampling_ based on hard LSH collisions to approximate the angular kernel. RACE Attention instead uses a _smooth, differentiable relaxation_ of RACE hashing ([2], [3]) to approximate the angular kernel. Note that YOSO’s training cost scales quadratically in $d$, whereas RACE’s training cost is linear in $d$.
>
> **2.) Theoretical guarantees:** YOSO provides no formal analysis of approximation error. In contrast, RACE comes with _explicit bias and variance guarantees_ (Theorem 2) and proves convergence to the corresponding $P$-powered angular kernel, giving a principled understanding of estimator quality.
>
> **3.) Applicability and scalability:** YOSO reports results only up to 4K sequence length and only for bidirectional attention. On the other hand, RACE supports full and causal attention, demonstrating bidirectional attention up to 64K tokens, and evaluating perplexity on the widely used WikiText-103 benchmark for causal language modeling.
>
> **For an in-depth comparison with YOSO please refer to answer to Q6 of Reviewer if3U.**
>
> **[1] Zhanpeng Zeng, Yunyang Xiong, Sathya N. Ravi, Shailesh Acharya, Glenn Fung, Vikas Singh. You Only Sample (Almost) Once: Linear Cost Self-Attention Via Bernoulli Sampling. ICML 2021.**
>
> **[2] Benjamin Coleman, Anshumali Shrivastava. Sub-linear RACE Sketches for Approximate Kernel Density Estimation on Streaming Data. WWW 2020.**
>
> **[3] Benjamin Coleman, Richard G Baraniuk, Anshumali Shrivastava. Sub-linear Memory Sketches for Near Neighbor Search on Streaming Data. ICML 2020.**
>
> > Q2. Reporting additional experiments.
>
> A2. We appreciate the reviewer for the emphasis on longer-context evaluation. In our initial submission, we limited experiments to sequences up to 8K tokens because, based on our theoretical analysis and earlier pilot studies, we did not anticipate any qualitative change in RACE Attention’s behavior at longer contexts, as the underlying hashing and aggregation mechanism is expected to behave consistently as sequence length grows. However, to more comprehensively validate this and to strengthen the empirical performance as reviewer W9FA suggested, we now extend our experiments to substantially longer sequence lengths. Specifically, in the revised paper, we include accuracy results for experiments at **16K**, **32K**, and **64K** sequence lengths on the ArXiv classification task in Table 7. In addition, we provide long-context image classification experiment on Food-101 dataset using Vision Transformers at **16K**  sequence length in Table 9. These new results (summarized below) confirm our original expectation: **RACE Attention maintains stable accuracy and continues to scale efficiently even as context length increases by an order of magnitude.**
>
> **Table:** Long-context ArXiv text classification performance on a 40GB A100 GPU. Train/Test denote per-epoch runtimes in seconds, and Acc. denotes test accuracy.
> | **Method**               |      |    **16K**      |          |    |      |      **32K**     |          |    |       |    **64K**      |          |
> |--------------------------|------------|----------|----------|----|-------------|----------|----------|----|-------------|----------|----------|
> |                          | **Train ↓**| **Test ↓** | **Acc. ↑** | │  | **Train ↓** | **Test ↓** | **Acc. ↑** | │  | **Train ↓** | **Test ↓** | **Acc. ↑** |
> | RACE (P=2,L=2)           | **80.5s**  | 3.9s     | 70.3%     | │  | **282s**    | 15s      | 89.4%     | │  | **561s**    | 22s      | 97.14%    |
> | RACE (P=3,L=3)           | 82.4s      | 4.0s     | **71.3%**     | │  | 289s        | 15.6s    | 90.6%     | │  | 584s        | 22.5s    | **97.92%**    |
> | RACE (P=4,L=4)           | 84.7s      | 4.1s     | 70.8%     | │  | 305s        | 16s      | **91.1%**     | │  | 594s        | 22.9s    | 97.45%    |
> | Linear                   | 83.8s      | 4.0s     | 67.9%     | │  | 286s        | 15.9s    | 87.3%     | │  | 591s        | 22.8s    | 96.35%    |
> | Linformer-128            | 86s        | **3.2s** | 64.1%     | │  | 296s        | **10.7s**| 87.5%     | │  | 616s        | **15.2s**| 97.4%     |
> | Performer-256            | 128s       | 5.8s     | 68.9%     | │  | 449s        | 24.6s    | 86.5%     | │  | 952s        | 35s      | 96.61%    |
> | FlashAttention2          | 95.7s      | 3.7s     | 69.8%     | │  | 471s        | 20s      | 89.7%     | │  | 1645s       | 47s      | 97%       |

---

> ### Author Response · Authors · 2025-11-20
> **Response (2/2) to Reviewer W9FA**
>
> > Q2. Reporting additional experiments (continued...)
>
> **Table**: Long-context (16K) image classification (ViT) performance on a 40GB A100 GPU with Food-101 dataset. Train/Test denote per-epoch runtimes in seconds, and Acc. denotes test accuracy. *Linear Attention, Linformer, and Performer use batch size = 1 due to OOM at batch size = 8. RACE and FlashAttention2 remain memory-efficient and use batch size = 8.*
>
> | **Method**            | **Train ↓** | **Test ↓** | **Acc. ↑** |
> |-----------------------|-------------|------------|------------|
> | RACE (P=2, L=2)   | **891s**    | **37s**    | 42.4%      |
> | RACE (P=3, L=3)       | 950s        | 40s        | **43.5%**  |
> | RACE (P=4, L=4)       | 1042s       | 42s        | 40.3%      |
> | Linear                | 1166s       | 44s        | 41.4%      |
> | Linformer-128         | 1250s       | 49s        | 20.2%      |
> | Performer-256         | 2546s       | 105s       | 42.4%      |
> | FlashAttention2       | 2600s       | 95s        | 42.1%      |
>
> We add the aforementioned table in the updated paper as Table 9 in Appendix.
>
> > Q3. For fair comparison, do efficiency results need accuracy context and vice-versa?
>
> A3. We thank the reviewer for raising this point. Our efficiency experiments follow well-established practice in prior work on scalable attention mechanisms, including HyperAttention [5] (see fig. 4 in [5]) and Performer [6] (see fig. 3 in [6]), and we take care to avoid hyperparameter tuning that would artificially favor runtime or memory measurements.
>
> In our accuracy evaluations, for RACE Attention, we fix the hyperparameters across context lengths 128-64K, i.e., $P,L \in$ {$2,3,4,5$}. The base configuration is identical in all methods per task (see Table 8). In the context of scaling experiments, for RACE we use the same $P,L \in$ {$2,3,4,5$}, and use identical configuration across all methods (e.g., embedding dimension, number of heads, batch size). We do _not_ re-tune any method to optimize speed or memory.
>
> While linear attentions can appear arbitrarily efficient by collapsing feature dimensions (e.g., setting the feature dimension to 1), such configurations severely degrade accuracy and are not used in meaningful applications. To illustrate this, we trained the Linear Attention baseline on CIFAR-10 dataset with feature dimension 10 and observed a drop in accuracy from $\sim 60$% to $\sim 50$%. These degenerate settings are exactly what our fixed-hyperparameter protocol is designed to avoid. Moreover, RACE Attention scales linearly with embedding dimension $d$, whereas Linear Attention scales quadratically in $d$, further justifying the controlled comparison.
>
> Under these accuracy-preserving settings, RACE Attention performs comparably to linear attentions for context lengths up to 64K. The meaningful difference in efficiency emerges beyond 128K tokens, where RACE continues to scale stably while many existing methods become infeasible. Training full models at 2-4M token context lengths is prohibitive due to memory and runtime limits, and our goal is to highlight this ultra-long-context regime, where RACE remains tractable _without any hyperparameter tuning_.
>
> Finally, due to limited compute resources, training full models at extremely long contexts (128K+) is challenging universally. Nevertheless, our aim is twofold: (1) to rethink attention mechanisms for extreme long-context training, and (2) to demonstrate that RACE can operate in a regime where other attention mechanisms cannot. For these reasons, we believe the efficiency numbers in figs. 4-6 reliably reflect the practical scalability of RACE. We appreciate the opportunity to clarify this point. We explicitly note the hyperparameters of RACE in captions of figs. 4–6, which are same across all accuracy evaluations.
>
>
> **[5] Insu Han, Rajesh Jayaram, Amin Karbasi, Vahab Mirrokni, David P. Woodruff, Amir Zandieh. HyperAttention: Long-Context Attention in Near-Linear Time. ICLR 2024.**
>
> **[6] Krzysztof Choromanski, Valerii Likhosherstov, David Dohan, Xingyou Song, Andreea Gane, Tamas Sarlos, Peter Hawkins, Jared Davis, Afroz Mohiuddin, Lukasz Kaiser, David Belanger, Lucy Colwell, Adrian Weller. Rethinking Attention with Performers. ICLR 2021.**

---

### Author Response · Authors · 2025-11-20
**Response to all Reviewers**

First of all, we would like to thank all the Reviewers for carefully reading our paper and for their insightful comments. We have updated the paper to incorporate the following revisions according to Reviewers' feedback:

- **Discussion about YOSO Attention:** We outline the key differences and provide an in-depth comparison between the estimators of RACE and YOSO [1] on lines 51-59 and 263-272. They both use LSH, but their mechanisms differ substantially. While YOSO uses hard Bernoulli sampling from collisions, RACE uses a novel, differentiable estimator of a $P$-powered angular kernel ([2], [3]). Secondly, YOSO provides no formal error bounds. In contrast, RACE offers explicit bias/variance guarantees and convergence analysis (Theorem 2). Finally, YOSO's experiments are limited to  bidirectional attention up to 4K tokens, while RACE provides accuracy up to 64K tokens, supports full/causal attention, and evaluates causal language modeling task on WikiText-103 (standard benchmark). Thus, RACE offers a distinct kernel framework, stronger theory, and broader applicability. For a detailed explanation, we refer the reader to our response to Q6 of Reviewer if3U.

- **Additional long-context text classification and image classification experiments:** We conduct extensive experiments on substantially longer contexts to re-validate the efficiency and performance of RACE Attention. Specifically, we report results at **16K**, **32K**, and **64K** sequence lengths on the ArXiv classification task, including both training and inference time per epoch on a single A100 GPU. We additionally evaluate long-context image classification on Food-101 dataset using Vision Transformers at **16K** sequence length. Across all settings, RACE Attention matches or outperforms the baselines while delivering faster runtime. Please refer to Tables 7, 9.

- **Scaling experiments with Sigmoid Attention and FlashAttention3 on GH200 (96 GB):** We add two new baselines to our scaling experiments as suggested by Reviewer eQBU. The efficiency of RACE is clear: **CPU-RACE is up to 20x faster, and GPU-RACE is up to 2500x faster than FlashAttention-3 at 4M context length.** Please refer to Tables 5, 6, 7, and 8.

- **Significance of the Scaling experiments:** To clarify unambiguously, our scaling experiments stress-test only the attention mechanism itself, not end-to-end training of a full Transformer. This follows well-established practice in prior scalable-attention works such as HyperAttention [4] and Performer [5]. This decomposition is intentional: at extreme context lengths, attention dominates both memory and compute, and many alternative mechanisms fail well before reaching this regime. Demonstrating that attention alone can scale to tens of millions of tokens is therefore a necessary prerequisite for training full models at such lengths. Note that our stress tests benchmark a single forward–backward pass of a multi-head attention layer under fixed hyperparameters **(identical to those used in our accuracy evaluations)**, ensuring no hyperparameter re-tuning is done to favor speed or memory.

- **Polishing the paper:** We clarify how the feature maps $\phi(Q_i)$, $\phi(K_j)$ relate to Angular Attention and explain the role of the softmax operation in Algorithm 1 (lines 300-312). Additionally, we include fig. 2 to illustrate how a modest increase in the sharpening parameter $\gamma$ in Angular Attention can closely mimic Softmax Attention. Finally, we add a concluding paragraph outlining future directions for RACE Attention, and following Reviewer if3U's suggestion, we have revised the paper’s title to **"RACE Attention: A Linear-Time Attention Mechanism for Long-Sequence Training with Extreme-Length Attention-Layer Scaling"** which reflects the main contributions.

Please refer to the individual reviewer responses for detailed explanations. Each reviewer response has its own set of references.

**References:**

**[1] Zhanpeng Zeng, Yunyang Xiong, Sathya N. Ravi, Shailesh Acharya, Glenn Fung, Vikas Singh. You Only Sample (Almost) Once: Linear Cost Self-Attention Via Bernoulli Sampling. ICML 2021.**

**[2] Benjamin Coleman, Anshumali Shrivastava. Sub-linear RACE Sketches for Approximate Kernel Density Estimation on Streaming Data. WWW 2020.**

**[3] Benjamin Coleman, Richard G Baraniuk, Anshumali Shrivastava. Sub-linear Memory Sketches for Near Neighbor Search on Streaming Data. ICML 2020.**

**[4] Insu Han, Rajesh Jayaram, Amin Karbasi, Vahab Mirrokni, David P. Woodruff, Amir Zandieh. HyperAttention: Long-Context Attention in Near-Linear Time. ICLR 2024.**

**[5] Krzysztof Choromanski, Valerii Likhosherstov, David Dohan, Xingyou Song, Andreea Gane, Tamas Sarlos, Peter Hawkins, Jared Davis, Afroz Mohiuddin, Lukasz Kaiser, David Belanger, Lucy Colwell, Adrian Weller. Rethinking Attention with Performers. ICLR 2021.**

---

### Author Response · Authors · 2025-11-30
**Summary for Area Chair(s): Reviewers' Concerns and Our Clarifications**

Given the recent OpenReview incident and the interruption of the discussion phase, we would like to provide a concise summary of the review process for our submission. Our intention is to offer a clear overview of the reviewers’ concerns and how we addressed all of them during the rebuttal period. Finally, we highlight the novelty of our work, emphasizing its contributions to scalable and accurate long-context attention.

### **Reviewer W9FA:**

**1. Comparison with YOSO**: The reviewer noted that our original submission did not sufficiently explain how RACE Attention differs from YOSO, despite shared motivation around LSH-based angular kernels. In response, we substantially expanded the discussion (lines 51–59 and 263–272), clarifying that YOSO relies on non-differentiable hard LSH, surrogate gradients, and estimates only the unnormalized kernel numerator via Bernoulli collisions and then performs post-hoc normalization. RACE instead uses a smooth, differentiable relaxation of RACE hashing to approximate both numerator and denominator with formal bias–variance guarantees. We also highlighted that RACE supports both causal and bidirectional attention and is evaluated up to 64K tokens, whereas YOSO reports results only up to 4K, positioning RACE as a more stable and scalable refinement.

**2. Experiments Beyond 8K Contexts:** The reviewer requested evaluation at longer contexts. We extended experiments to 16K, 32K, and 64K variants on ArXiv classification (Table 7) and added 16K long-context ViT experiments (Table 9). All methods were run with matched hyperparameters to ensure clean scaling comparisons. These results demonstrate that RACE maintains accuracy while scaling strictly linearly.

**3. Accuracy–Efficiency Tradeoffs:** The reviewer emphasized that efficiency plots can be misleading if methods use degenerate hyperparameters. We clarified that all scaling experiments follow accuracy-preserving configurations consistent with existing long-context literature (e.g., HyperAttention, Performer). Hyperparameters were held fixed across all methods and context lengths, and we explicitly documented these settings in captions. Under these controlled settings, RACE matches baselines up to 64K and remains scalable at extreme lengths where others run out of memory.

### **Reviewer eQBU:**

**1. Additional Baselines:** The reviewer requested inclusion of FlashAttention3 and Sigmoid Attention. We added both baselines to the accuracy and scaling comparisons, validating that RACE Attention maintains stable accuracy while continuing to scale efficiently even as the context length increases.

**2. Softmax in Algorithm 1:** The reviewer was concerned that Algorithm 1 appeared to reintroduce softmax. We clarified that the operation is not the Transformer softmax but a local smoothing machinery ensuring differentiable LSH bucketization.

**3. Connection Between Feature Maps and Angular Attention:** We added a complete derivation showing how our soft feature map arises from the $P$-powered angular kernel via LSH and RACE sketches, including collision probabilities and propagation to the output in lines 300-313.

**4. Understanding the Sensitivity of the Sharpening Parameter $\gamma$:** We clarified that $\gamma$ is not tunable but fixed by the number of hyperplanes ($\gamma=P$). We also added a Frobenius-norm analysis showing that approximation error $||angular–softmax||_F$ decreases as $\gamma$ increases, and that accurate approximations require only modest polynomial degrees in Fig. 2.

### **Reviewer if3U:**

**1. Title Framing and Interpretation of Scaling Experiments:** The reviewer sought clarity on whether the 75M token results reflect full-model capability or single-layer stress tests. We emphasized that these results reflect one attention layer’s forward-backward pass. To avoid misinterpretation, we revised the title to more precisely reflect the work’s scope: a linear-time attention mechanism scalable to extreme-length sequences, not full-model context windows.

**2. Causal Masking:** This was a minor clarification from the reviewer regarding whether causal masking is straightforward to incorporate. We explained that while we were able to implement the causal RACE Algorithm (Algorithm 2), the full theoretical analysis is non-trivial and outside the scope of this paper. For completeness, we refer the AC(s) to our detailed response to Q4 for reviewer if3U.

**3. YOSO Comparison:** We already outline this above for reviewer W9FA.

### **Novelty of Our Work**
RACE Attention introduces the first differentiable sketch for angular-kernel attention, enabling end-to-end training while scaling to unprecedented context lengths (75M tokens on CPU, 12M on GPU) under accuracy-preserving settings. Beyond this, RACE offers formal approximation guarantees, efficient CPU/GPU kernels for causal and non-causal training, and extensive ablations that demonstrate stable accuracy alongside linear runtime and memory complexity.

---

### Author Response · Authors · 2025-12-03
**Paper Revision Summary**

We thank all the reviewers and the area chair(s) for their time and constructive feedback in helping us improve the paper. We are encouraged that reviewers found the problem both important and timely (R **W9FA**, R **eQBU**, R **if3U**), appreciated the strength of our scaling experiments, the systems-level implementation, and the rigor of our theoretical analysis (R **eQBU**, R **if3U**). We are also grateful that reviewers highlighted the algorithm’s broad applicability, simple implementation, and its potential usefulness to the community (R **W9FA**, R **if3U**).

We have updated the paper to address all reviewers' concerns. The major revisions are summarized below:

**1. [W9FA, if3U]:** clarified the technical and theoretical novelty of RACE relative to YOSO in lines 51–59 and 263–272.
**2. [W9FA]:** expanded the evaluation to sequence lengths $8\times$ larger than the original experiments, as shown in Tables 7 and 9 for text and image classification.
**3. [W9FA]:** explained the correlation between efficiency and accuracy by stating that the scaling experiments employ exactly the same hyperparameters used in the accuracy experiments, as detailed in Figures 4, 5, and 6.
**4. [eQBU]:** added FlashAttention-3 and Sigmoid Attention as additional baselines in both the accuracy and scaling tables.
**5. [eQBU]:** expanded the explanation of the softmax operation in Algorithm 1 and articulated the linkage between soft feature maps $(\phi(Q_i), \phi(K_j))$ and angular attention in lines 300–313.
**6. [eQBU]:** included a new plot demonstrating that the Frobenius error between the angular and softmax formulations decreases with increasing $\gamma$ (Figure 2).
**7. [if3U]:** revised the title to better capture the core contributions of this work: strictly linear complexity in sequence length $N$ and dimension $d$, and the ability to scale the attention primitive to extreme sequence lengths.
**8. [if3U]:** added the causal masking variant of RACE (Algorithm 2) to the Appendix to illustrate its implementation.

**For a detailed summary of the full discussion, please refer to the next response.**

---

### Meta-Review · Area_Chair_u14c · 2026-01-06

**Summary:**

Initial reviews for the paper are somewhat mixed (2,4,6).

Reviewer W9FA notes that the work has similarities to existing work and notes that experiments have only been done on relatively short sequences, when the main pitch of the work is to enable very long context length.  Additionally, there are questions about the accuracy/computation trade-off based on how the embedding dimension is selected.

Reviewer eQBU raises questions about whether the most recent implementations of FlashAttention were using in the benchmark comparisons and also asks about alternative kernels (such as Sigmoid attention) which might allow for simple kernel implementations.  There are also questions to clarify certain aspects of the presentation.

Reviewer if3U also notes similarities to existing work and questions whether some of the claims are somewhat oversold in that the maximum computable context length reported in the abstract (75 million tokens) refers to an attention primitive and not a full model.

**Reviewer Concerns:**

The majority of the concerns appear largely addressed in the rebuttal, with perhaps the exception of the framing of the work with respect to prior work as discussed below.  Reviewer eQBU explicitly notes all of their concerns being addressed and raising their score.

With regards to connections with prior work, the authors have provided fairly detailed responses about how their work differs from this prior work and how it results in practical benefits for scaling in the embedding dimension and making the overall operator differentiable.  Likewise, they have made corresponding modifications to the manuscript to delineate the differences between the two works.  In my view, the authors have largely addressed differentiating the current work from prior work.

**Reviewer Scores:**

The review of this paper would have benefited significantly from a more substantial discussion period.

Reviewer eQBU notes that their concerns have been addressed and raised their score, while reviewer if3U notes that their support of the paper decreased after it was clarified that the 75M token context length referred to a single attention primitive and not a full model along with concerns of over-selling the work and it's novelty to prior work.  However, in response to this the authors have provided a more substantial discussion of how their work differs from this prior work along with corresponding modifications to the manuscript, yet reviewer if3U was not able to respond to this comment before the discussion was closed.

Whether reviewers eQBU and if3U would be satisfied with the authors' final response and paper modifications I cannot say, but in my view the authors have provided a strong discussion of the differences with the noted prior work and how this results in several practical advantages for the ultimate algorithm.  This, combined with several other positive aspects of the work noted by the reviewers regarding the systems work and experimental demonstration, leave me inclined to give the authors the benefit of the doubt and recommend acceptance for the paper.  Nevertheless, I would encourage the authors to be very judicious how they frame their work.  For example, even in the current abstract it is still somewhat ambiguous that the 75M token capacity refers to an attention primitive and not a full model.

---

### Decision · Program_Chairs · 2026-01-26

Accept (Poster)